

# Large second-order Josephson effect in planar superconductor-semiconductor junctions

P. Zhang[1]°, A. Zarassi[1], L. Jarjat[2], V. Van de Sande[3], M. Pendharkar[4],
J. S. Lee[5], C. P. Dempsey[4], A. P. McFadden[4], S. D. Harrington[6], J. T. Dong[6],
H. Wu[1], A. -H. Chen[7], M. Hocevar[7], C. J. Palmstrøm[4,5,6] and S. M. Frolov[1]⋆

**1** Department of Physics and Astronomy, University of Pittsburgh, Pittsburgh, PA, 15260, USA
**2** Département de Physique, Ecole Normale Supérieure, 75005 Paris, France
**3** Eindhoven University of Technology, 5600 MB, Eindhoven, The Netherlands
**4** Electrical and Computer Engineering, University of California Santa Barbara,
Santa Barbara, CA 93106, USA
**5** California NanoSystems Institute, University of California Santa Barbara,
Santa Barbara, CA 93106, USA
**6** Materials Department, University of California Santa Barbara,
Santa Barbara, CA 93106, USA
**7** Univ. Grenoble Alpes, CNRS, Grenoble INP, Institut Néel, 38000 Grenoble, France

⋆ frolovsm@pitt.edu

## Abstract

We investigate the current-phase relations of Al/InAs-quantum well planar Josephson junctions fabricated using nanowire shadowing technique. Based on several experiments, we conclude that the junctions exhibit an unusually large second-order Josephson harmonic, the $\sin(2\varphi)$ term. First, superconducting quantum interference devices (dc-SQUIDs) show half-periodic oscillations, tunable by gate voltages as well as magnetic flux. Second, Josephson junction devices exhibit kinks near half-flux quantum in supercurrent diffraction patterns. Third, half-integer Shapiro steps are present in the junctions. Similar phenomena are observed in Sn/InAs quantum well devices. We perform data fitting to a numerical model with a two-component current phase relation. Analysis including a loop inductance suggests that the sign of the second harmonic term is negative. The microscopic origins of the observed effect remain to be understood. We consider alternative explanations which can account for some but not all of the evidence.

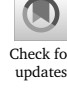

## Contents

°Current address: Beijing Academy of Quantum Information Sciences, 100193 Beijing, China.



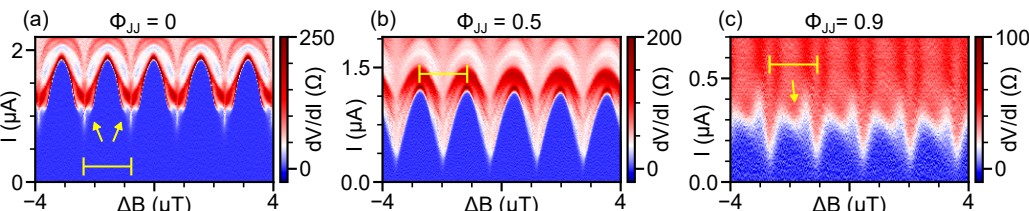

Figure 1: (a) Schematic diagram of SQUID-1. The loop consists of two planar SNS Josephson junctions, $JJ_a$ and $JJ_b$. The magnetic field ($B$) is perpendicular to the device plane. (b) Optical image of the device. The area labeled "Al/2DEG" is the superconducting mesa. (c) Switching current ($I_{sw}$) as a function of applied gate voltage ($V_g$) on $JJ_a$ or $JJ_b$. One junction is tuned to the non-superconducting regime when measuring the other one. (d) Differential resistance ($dV/dI$) as a function of current ($I$) and magnetic field. $\Phi_{JJ}$ is the nominal magnetic flux extracted from the Fraunhofer-like background. Gate voltages $V_{g,a} = V_{g,b} = 500$ mV. In panels (c) and (d) field is offset by -0.245 mT.

# 1 Broad context

Growing interest in the integration of new materials into quantum devices offers opportunities to study proximity effects, such as between superconductors and semiconductors. Elements as basic as planar junctions, with two superconductors placed side-by-side, can host Majorana zero modes, and be used as nonlinear quantum circuit elements [1–3]. Depending on the junction material, planar junctions can also be used to search for exotic phenomena such as triplet superconductivity [4].

# 2 Background: Current phase relations

The primary characteristic of a Josephson junction (JJ) is the current phase relation (CPR) [5, 6]. A CPR connects the supercurrent to the phase difference across the junction and is calculated from the weak link energy spectrum and interface transparency. Among other properties, it predicts the electromagnetic response of a Josephson junction in a circuit.

The CPR of a superconductor-insulator-superconductor (SIS) Josephson junction, as originally derived, is sinusoidal, $I(\varphi) = I_c \sin \varphi$, where $I$ is the supercurrent, $\varphi$ is the phase difference between two superconducting leads, and $I_c$ is the critical current. More generally, a CPR can be decomposed into a Fourier series, $I(\varphi) = \sum_n I_n \sin(n\varphi)$ [6]. In real SIS devices, higher-

Figure 2: SQUID-1 oscillations show additional modulation. (a-c) Differential resistance ($dV/dI$) as a function of the current and the magnetic field near $\Phi_{JJ} = 0$, 0.5 and 0.9, respectively. Horizontal scale bars indicate the primary period of SQUID oscillations. Arrows in (a) and (c) show additional kinks in the oscillation. $V_{g,a} = 128$ mV, $V_{g,b} = 113$ mV, chosen to set switching currents to 0.9 $\mu$A in both junctions.

order components may be present but the amplitude would be small [7]. In superconductor-normal metal-superconductor (SNS) JJs, the CPR can deviate significantly from the sinusoidal form due to contributions from Andreev bound states which are subgap quasiparticle states.

In the clean SNS limit, the CPR is predicted to be linear or a skewed sine function [8–17]. Even in junctions with skewed CPR, the first harmonic, $\sin(\varphi)$ is typically dominating. $4\pi$-periodic, or the half-integer harmonic in CPR is predicted in topological superconductors and explored in a variety of materials [18, 19].

# 3 Previous work: Second order Josephson effect

Large second-harmonic CPR was searched for in a variety of junctions at the so-called 0-$\pi$ transition [20–25] or in 45°-twisted high-temperature superconductor junctions [26–30]. In these cases the second harmonic can be observed because the first harmonic is cancelled.

In SNS junctions all harmonics are present, including the second harmonic. However because of the presence of even higher-order terms, we do not expect double modulation. If a SQUID consisting of such junctions is flux-biased to $\pi$, the first harmonic can be canceled, resulting in the domination of the second harmonic signatures [31–35]. This effect is used to create the $\pi$-periodic energy-phase relation for so-called "$0-\pi$ qubit" [33, 36–38].

A possible experimental signature of the second-order Josephson effect is half-integer Shapiro steps [39], which can also have a variety of other origins due to phase locking of Josephson vortices or quasiparticle dynamics [40, 41]. Another type of evidence is in super-current interference patterns where additional minima or kinks are observed at values of half-integer magnetic flux quanta [20, 25].

# 4 List of results

In our experiments, we find Josephson junctions with an unusually large second harmonic in planar junctions based on an InAs quantum well. We find confirmation of this in several measurements. First, we observe that the superconducting quantum interference devices (SQUIDs) made of two planar InAs junctions exhibit extra kinks in the flux modulation. The shape of the SQUID characteristics evolves with junction asymmetry, which can be explained by a simple two-component CPR model. Second, single planar junction supercurrent diffraction patterns show kinks near half-quantum of applied flux. Finally, half-integer Shapiro steps are also observed. In our single junction measurements, the observation of the large $\sin(2\varphi)$ term is not related to a cancellation of the first-order term, e.g. at the 0-$\pi$ transition.

The amplitude of the second harmonic from several of these measurements is found to be around 0.4 of the first harmonic. The sign of the second harmonic is determined to be negative from the model that includes the loop inductance (see supplementary information). Negative sign is expected upon decomposition of a skewed sinusoidal function into Fourier components. Because of the long mean free path on the order of the junction length and high interface transparency, it is not surprising if the CPR has higher-order terms. However, a skewed function should contain all sinusoidal harmonics, including those higher than two. We do not observe an apparent third-order or higher-order terms, which is surprising and requires further studies.

## 5 Brief methods

Planar InAs quantum well junctions are prepared using the nanowire shadow method [42,43]. Because the junction area is not subjected to chemical or mechanical etching, this approach preserves the quantum well in the junction area. Superconductors used are Al or Sn. We focus on Al devices in the main text. Data from Sn devices are available in supplementary materials.[1] Standard electron beam lithography and wet etching is used to pattern the mesa outside the junction. Measurements are performed in a dilution refrigerator at 50 mK unless otherwise stated.

## 6 Figure 1 description

We first present examples of dc-SQUID patterns with extra features that we study in the context of the $\sin(2\varphi)$ terms. Device SQUID-1 is depicted schematically in Fig. 1(a) and an optical microscope image is shown in Fig. 1(b). The device has two Josephson junctions in parallel, $JJ_a$ and $JJ_b$. Two nanowire top-gates $V_{g,a}$ and $V_{g,b}$ are used for tuning critical currents. We set supercurrent to zero in one junction with its gate and measure the gate dependence of the switching current $I_{sw}$ in the other junction [Fig. 1(c)]. $I_{sw}$ is typically referred to as critical current, though the true Josephson critical current can be higher than the switching current. $I_{sw}$ quenches at gate voltages near $-100$ mV and saturates above 300 mV in both junctions. Both junctions have the same nominal geometry, so their magnetic flux modulation, or diffraction patterns are expected to have similar periods. This can be confirmed in Fig. 1(d), where there is only one Fraunhofer-like low-frequency modulation while both junctions are in the superconducting state. With the Fraunhofer period of 0.29 mT and the junction width of 5 $\mu$m, we get an effective junction length of 1.4 $\mu$m (an order of magnitude larger than the typical physical length) for both JJs. The long effective length is likely due to large London penetration depth which is typical for thin film superconductors. We use the junction magnetic flux $\Phi_{JJ}$, given in the unit of $\Phi_0 = h/2e$. The SQUID flux and the junction flux are applied from a large superconducting magnet and cannot be independently controlled. The high frequency oscillations within the Fraunhofer-like envelope are due to interference between the two junctions. The period 1.57 $\mu$T gives a SQUID area of $1.31 \times 10^3$ $\mu$m$^2$ which is similar to the area of the inner loop ($1.28 \times 10^3$ $\mu$m$^2$). The oscillations are shown in detail in Fig. 2.

## 7 Figure 2 description

SQUID-1 modulation patterns are shown in Figs. 2(a)-2(c) for different junction flux $\Phi_{JJ}$. Two junctions are tuned to have the same zero-field switching current so that the SQUID is symmetric ($I_{sw,a} = I_{sw,b} = 0.9$ $\mu$A). At $\Phi_{JJ} = 0$, the pattern exhibits kinks near SQUID oscillation minima [Fig. 2(a) yellow arrows, zoomed-in data in Fig. 15]. We also observe that, despite tuning the SQUID to the symmetric point, the nodes of the modulation patterns are lifted from zero. Pattern at $\Phi_{JJ} = 0.5$ [Fig. 2(b)] does not contain extra features. Here, $I_{sw}$ follows a $|\cos B|$ curve as expected for standard symmetric SQUID without high-order harmonics in its CPR. When $\Phi_{JJ}$ is 0.9, the pattern has a distinct double-modulation character. Extra minima in $I_{sw}$ are observed near half-period of SQUID modulation [Fig. 2(c) yellow arrow].

---

[1]See supplementary materials for detailed models, extended data, and more discussion.

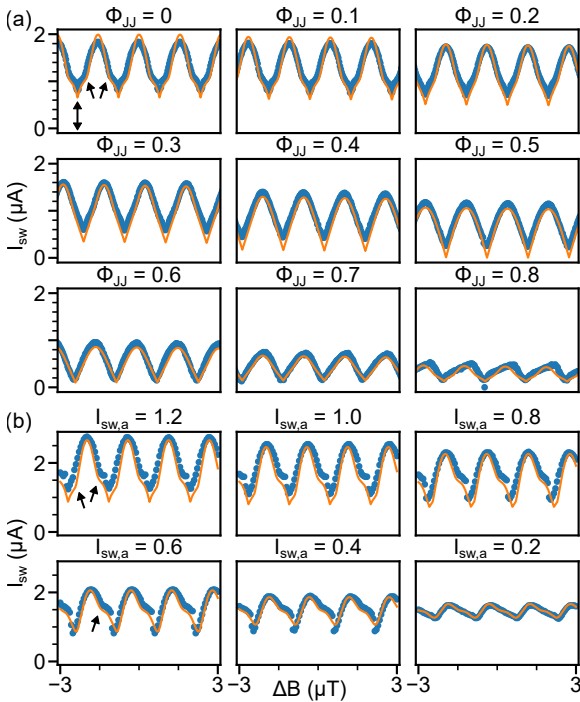

Figure 3: Extracted switching current (blue circle) and simulated critical current (orange line) in SQUID-1. (a) SQUID oscillations at a variety of $\Phi_{JJ}$. The device is tuned to a symmetric state where $I_{sw,a} = I_{sw,b} = 0.9$ $\mu$A. (b) SQUID oscillations for a variety of $I_{sw,a}$. Here, $\Phi_{JJ} = 0$ and $I_{sw,b} = 1.47$ $\mu$A. Arrows are discussed in the text. Numerical offsets of order 0.1 mT are applied to the magnetic field to compensate for trapped magnetic flux.

# 8 Figure 3 description

Next, we extract $I_{sw}(\Delta B)$ traces from data like in Fig. 2 and fit them to a basic model for a SQUID with two-harmonic CPR junctions $I(\varphi) = I_1 \sin(\varphi) + I_2 \sin(2\varphi)$ (see supplementary materials for model description and detailed parameters[1]). The numerical model uses $I_2/I_1 = 0.4$ the ratio between the second and the first Josephson harmonics in the CPR, for both junctions a and b.

In Fig. 3(a), the device is tuned to a symmetric state where the zero-field switching currents in $JJ_a$ and $JJ_b$ are similar ($I_{sw,a} = I_{sw,b} = 0.9$ $\mu$A). As the junction flux $\Phi_{JJ}$ increases, the amplitude of $I_{sw}$ shifts towards lower values due to the Fraunhofer-like envelope. The model reproduces the basic features. Kinks, pointed out by single-ended arrows in panel (a), are most apparent near junction flux $\Phi_{JJ} = 0$ (zoomed-in data in Fig. 15). Node lifting (double arrows) is also reproduced by the model. While it is typically associated with asymmetric SQUIDs, in this model it originates from the second-order Josephson effect. Additional features disappear both in the data and in the model near $\Phi_{JJ} = 0.5$, where the pattern closely follows the standard $|\cos B|$ SQUID relation. When $\Phi_{JJ}$ approaches 0.8 the experimental curve becomes skewed while the simulated curve is more symmetric.

In Fig. 3(b) SQUID-1 is tuned by gates, and is asymmetric while $\Phi_{JJ}$ is 0. $I_{sw,b}$ is fixed at 1.47 $\mu$A while $I_{sw,a}$ is changing. The amplitude of the oscillation decreases as $I_{sw,a}$ decreases. The oscillations are asymmetric. At $I_{sw,a} = 1.2$ $\mu$A, two kinks (arrows) are at different $I_{sw}$ heights. As $I_{sw,a}$ decreases, one kink becomes difficult to resolve. In this case skewed patterns are captured by the model, e.g. at $I_{sw,a} = 0.2$ $\mu$A. More examples of SQUID data are presented in supplementary materials from this device and additional SQUIDs.[1]

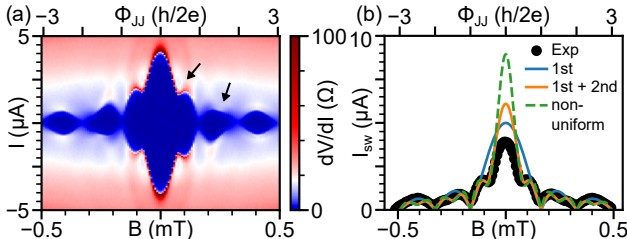

Figure 4: Supercurrent diffraction in single Josephson junction device JJ-1. (a) Differential resistance ($dV/dI$) as a function of the current ($I$) and the magnetic field ($B$). $I_{sw}$ manifests kinks near half-integer $\Phi_{JJ}$s (black arrows). Numerical off-sets of about 0.03 mT are applied to the magnetic field to compensate for trapped magnetic flux. (b) Extracted (black circle) and simulated $I_{sw}$ with models considering only the first harmonic ($I_1 = 5, I_2 = 0$, solid blue), first and second harmonics ($I_1 = 2.5, I_2 = 4.25$, solid orange) or only first harmonic but with piece-wisely distributed critical current density ($j_{side} = 5, j_{center} = 16.85$, dashed green).

## 9    Discussion of SQUID results

We first discuss the situation where $\Phi_{JJ} = 0$. Our SQUID model shows that for a symmetric SQUID at $\Phi_{JJ} = 0$, the nodes in $I_{sw}$ are lifted from 0 and two additional kinks appear as $I_2$ increases [Fig. 9(a)]. These two signatures are highlighted in Fig. 3(a). Thus the presence of additional kinks, and the node lifting are consistent with a significant second harmonic.

As the SQUID is made more asymmetric in Fig. 3(b), the SQUID modulation appears more skewed, meaning that the y-axis position of one kink decreases while the position of the other increases. When $I_{sw,a} << I_{sw,b}$ , $JJ_b$ passes a nearly fixed supercurrent (thus a constant phase difference) to maximize the total switching current $I_{sw}$ while the oscillation comes mostly from $JJ_a$. As a result, $I_{sw}(\Delta B)$ traces out the CPR curve of $JJ_a$ with a constant shift. This regime is reached for $I_{sw,a} = 0.6\,\mu$A and $0.4\,\mu$A. We notice that at $I_{sw,a} = 0.2\,\mu$A the kink is not resolved in the experimental curve but persists in the simulated curve. This may be due to a decrease in the junction transparency when $I_{sw,a}$ is small which decreases the amplitude of the second harmonic in the CPR, but can also be due to the decreased accuracy of extraction at lower currents.

Near $\Phi_{JJ} = 0.5$, $I_{sw}$ oscillates like a $|\cos B|$ function (Eq. D.4) and shows no kinks or extra minima. The device looks like an ordinary SQUID. This is because at this junction flux value the second harmonic within each junction is cancelled. While the first harmonic has diffraction minima at $\Phi_{JJ} = 1$, the second harmonic has nodes at $\Phi_{JJ} = 0.5$ where $I_2 = 0$.

The deviation from simulations at $\Phi_{JJ} = 0.8$ in Fig. 3(a) and $\Phi_{JJ} = 0.9$ in Fig. 2(c) can be explained by a small difference between two JJs' Fraunhofer periods. The actual flux is closer to 1 in one junction than the other when $\Phi_{JJ}$ is near 1. This difference makes the SQUID more and more asymmetric near $\Phi_{JJ} = 1$.

## 10    Figure 4 description

We also find second Josephson harmonic signatures in single junctions (Fig. 4). Similar results from other single JJs including those made from Sn/InAs 2DEG are available in Ref. [42].[1] The overall envelope $I_{sw}(B)$ is Fraunhofer-like, which is standard for single-harmonic uniform and sinusoidal junctions. However, in deviation from this behavior, kinks are observed at half-integer magnetic flux values [Fig. 4(a) black arrows]. The kink is most clear within the first

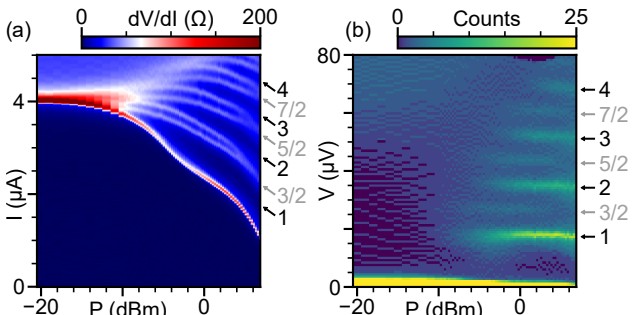

Figure 5: Half-integer Shapiro steps in JJ-1. (a) $dV/dI$ as a function of the current ($I$) and the microwave power ($P$) at zero magnetic field. The microwave frequency $f = 6.742$ GHz. Shapiro steps manifest as minima in $dV/dI$ (dark blue regimes between bright lines). (b) Histogram of voltage ($V$). Each Shapiro step becomes a maximum (bright yellow lines) in the histogram. Half-integer steps are visible. Shapiro indices are labeled on the right in both panels. Numerical offsets of about 0.03 mT are applied to the magnetic field to compensate for trapped magnetic flux.

lobe of the Fraunhofer-like modulation. The second lobe is skewed which is consistent with a minor kink. The pattern is not symmetric in positive-negative field but is inversion symmetric in field-current four-quadrant view. This is consistent with the self-field effect in these extended junctions.

We extract $I_{sw}$ and fit it with three different models in Fig. 4(b), detailed in supplementary materials.[1] The solid blue trace shows the basic Fraunhofer diffraction pattern for a single-component CPR. The two-component CPR model reproduces the kinks as shown in solid orange trace. Another model shown with dashed green line is for non-uniform critical current density and is discussed in the Alternative Explanations block.

Both non-uniformly distributed critical current and second harmonic in CPR produce half-period kinks. We notice that measured $I_{sw}$ of the single junction is smaller than the simulated value near zero field. This may be due to a non-uniformly distributed supercurrent or the fact that the experimentally measured switching current is smaller than the theoretical critical current [44]. Because the fit does not reproduce the data simultaneously in the central lobe and in the side lobes, we do not rely on the values of $I_2/I_1$ extracted from this analysis (see supplementary materials for details[1]). Half-periodic kinks were observed in anodic-oxidation Al/2DEG junctions but the analysis of the second-order Josephson effect has not been performed [45].

## 11 Figure 5 description

Shapiro steps in JJ-1 are presented at a microwave frequency of 6.742 GHz. A Shapiro step is a minimum in $dV/dI$ [Fig. 5(a)]. Another common form of plotting Shapiro step data is to bin data points by measured voltage across the junction [46]. In this form steps become straight bright lines in the histogram at Josephson voltages [Fig. 5(b)]. Apart from the usual integer steps, half-integer Shapiro steps are also observed at 3/2, 5/2 and 7/2. The missing step at 1/2 is likely due to self-heating [47–49].

While half-integer steps are clear and observed at relatively high frequencies, we do not observe steps at higher denominators such as 1/3 or 1/4. This is another confirmation that higher order Josephson harmonics, such as 3rd and 4th order, are not apparent in the data. Steps at 1/3 have been reported before in InSb nanowire junctions [50].

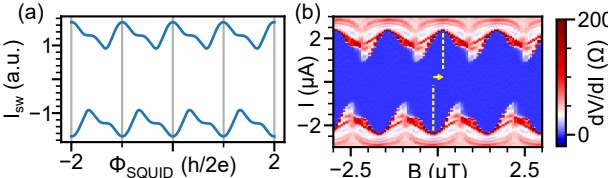

Figure 6: Asymmetries between negative and positive switching currents. (a) Simulated negative and positive switching currents. The loop inductance is not included in this model. Asymmetries arise due to the second-harmonic term in the current-phase relation. $I_{a,1}$, $I_{a,2}$, $I_{b,1}$, $I_{b,2}$ are 0.3, 0.15, 1, 0.5, respectively. (b) Experimental $dV/dI$ as a function of the current $I$ and magnetic field $B$, showing both negative and positive transitions in SQUID-1. The superconducting switchings qualitatively match the simulation in panel (a), except for a relative shift between negative and positive maximums (yellow arrow). The shift is due to the inductive effect which coexists with the second harmonic effect. $V_{g,a} = 100$ mV, $V_{g,b} = 500$ mV.

In these junctions we also observe missing integer steps at lower microwave frequencies, discussed in a separate manuscript [49]. This has been reported as a signature of the $4\pi$ Josephson effect characterized by the $\sin(\varphi/2)$ CPR. However, we observe the missing step pattern in the non-topological regime where fractional Josephson effect is not expected. We explain this through a combination of fine-tuning and low signal levels.

To illustrate that missing Shapiro steps are weak as evidence of unusual features in the CPR, we draw attention to the fact that the step at $n = 1/2$ is missing from our data. We do not take this as evidence of an "integer" Josephson effect.

## 12 Figure 6 discussion

Asymmetries between negative and positive switching currents in SQUIDs. The large second-order Josephson effect can produce asymmetries between positive and negative bias switching currents in junctions and SQUIDs [14, 34, 51–53]. These phenomena are heavily studied under the name "superconducting diode" in recent literature. To understand how the asymmetry arises we provide a simulation of our 2-junction 2-component CPR model in positive and negative bias (Figs. 6(a)). As can be seen, a kink in the positive bias is aligned with a dip in negative bias, and vice versa, an effect entirely due to the two-component CPR. This behavior is also present in the experiment (Fig. 6(b)). At the same time, the experimental data differ from the model in that the switching current maxima are not aligned for positive and negative bias in the experiment, but they are in the model. This difference is due to the finite inductance of the SQUID loop, of order 160 pH, resulting in finite phase winding due to circulating currents, an inductive effect [54, 55]. This effect provides another mechanism for the asymmetry. The simulation including both the inductance and the second Josephson harmonic shows that increasing the inductance would suppress signatures due to the second harmonic, e.g., the kinks at quarter flux values (Fig. 12).

Sign of the second harmonic term. Simulation results in the main text do not include the inductive effect, therefore they do not reveal the sign of the second harmonic term. This is because $I_{sw}$ is unchanged if both signs of the second harmonic term and the external flux are flipped.[1] The presence of finite SQUID inductance provides a simple way to determine the sign of the second harmonic term. The model combining both second harmonic and inductive effects shows that maxima in negative and positive $I_{sw}$s move towards kinks near half-integer flux values if the second-harmonic term is negative, and vice versa (Figs. 12(b) and 13(b)). The


SciPost Phys. **16**, 030 (2024)

movement of $I_{sw}$ maxima in the experiment (Fig. 6(b)) suggests a negative second harmonic term. This is expected if the second-order effect originates from high-transparency of the junction and from the skewed current-phase relation, which yields a negative second-order Fourier component. Note that the sign of the second-order term can also be determined by checking the directions of of the applied field and current flow.

## 13 Alternative explanations

Half-periodic kinks in a single JJ diffraction patterns can be explained by a non-uniformly distributed supercurrent. We reproduce these kinks with the model of a three-level supercurrent density distribution (Fig. 4(b) and supplementary materials[1]). The non-uniform supercurrent may be caused by resist residue introduced during fabrication. We observe stripes of residue in our first batch (Tab. 1, Al-chip-1) of junctions which includes JJ-1, caused by double electron beam exposure (Fig. 7). These residues are avoided and are not observed in newer devices without double exposure (Figs. 1(b),29(b),29(c)), which includes SQUID-1 and more devices in Fig. 29 and Ref. [42] - many of which do show additional modulation such as extra nodes and kinks.

It is worth noting that the non-uniform model produces a "lifted odd node" diffraction pattern that has been used to argue for the observation of the exotic fractional Josephson effect, characterized by $4\pi$-periodic current phase relations associated with Majorana modes [56]. We obtain the diffraction pattern of the same overall shape without the need to have this physics present [Fig. 4(b)]. In planar junctions it is difficult to know whether a node corresponds to integer or half-integer flux. This is because the junction area for supercurrent diffraction purposes can be much larger than the lithographic area. In our work, for instance, the junction length is close to 1.5 microns while the lithographic length is ~150 nm. This makes it hard to distinguish a half-integer kink from a lifted integer node.

Another aspect we consider is whether patterns such as those extracted in Fig. 3 or Fig. 4(b) are true representations of the switching current evolution. In some of the junctions the apparent extra modulation (blue area in colorplots, highlighted in Fig. 30(c) by changing the color) appears to be above the true switching current and may arise due to the evolution of finite-voltage resonances in magnetic field. One origin for these resonances is multiple Andreev reflections (MAR). See Figs. 21-24 for examples.

This type of artefacts does not explain all of our data. For example, in Fig. 4(a) we do not see MAR or faint finite-voltage state regions. Another argument we can give in support of the $\sin(2\varphi)$ origins of the SQUID patterns is the detailed agreement between our numerical model and the data. See supplementary materials for an extended discussion.[1]

Half-integer Shapiro steps have been attributed to effects not related to the second-order Josephson effect, such as a phase-locking of Josephson half-vortices to microwaves, and non-equilibrium quasiparticle dynamics [40, 41]. Hence their demonstration cannot by itself be used to claim the presence of $\sin(2\varphi)$ terms. We provide Shapiro data as additional rather than key evidence.

Taken together, we argue that the evidence from SQUIDs, single junctions and agreement with the model, are all consistent with a strong second-order Josephson effect. While alternative explanations can be used to support individual measurements, none of them can explain all of the data, though multiple factors may be present simultaneously with a lower likelihood.

## 14 Conclusion

In summary, we studied the current-phase relation in Josephson junctions and SQUIDs made with the nanowire shadow-mask method. Signatures due to a strong second-order harmonic in the CPR are observed. The simulation shows good agreement to experimental $I_{sw}$ data in the SQUID device suggesting a ratio of the second to first harmonics of $I_2/I_1 = -0.4$.

In the clean limit and at zero temperature, the CPR is a skewed $2\pi$-periodic function [6] with many higher order terms. After Fourier decomposition, one gets $|I_2/I_1| = 0.4$ and $|I_3/I_1| \approx 0.26$. So, in an ideal ballistic junction, the magnitude of the second harmonic we obtain would not necessarily be surprising. The surprising is the combination of doubly-modulated junction characteristics together with the lack of manifested higher order terms such as the third term. This indicates an unusual situation in which either the second order term is unusually large, or the higher order terms are suppressed through an undetermined mechanism.

## 15 Future work

The origin of such a strong second-order harmonic, not accompanied by even higher order terms (skewed CPR) still needs to be understood. The sensitivity of observed signatures to surface treatment can be studied. Higher harmonics can be used to engineer nonlinearity in quantum circuits.

## 16 Duration and volume of study

This study is divided into two periods, which correspond to period 2 and period 3 in Ref. [42].

The first period was between August 2018 to June 2019, including sample preparation, device fabrication and measurements. We explored the first-generation devices (Al, InSb nanowire, without gates). More than 7 devices on 1 chip are measured during 2 cooldowns in a dilution refrigerator, producing about 8900 datasets.

The second period was between March 2021 to February 2022. We explored the second-generation devices, (Al or Sn, InAs/HfO$_x$ nanowire, with self-aligned nanowire gates). 62 devices on 6 chips are measured during 8 cooldowns in dilution refrigerators, producing about 5700 datasets.

## 17 Data availability

Curated library of data extending beyond what is presented in the paper, as well as simulation and data processing code are available at [57].

## Acknowledgments

We thank J. Stenger, D. Pekker, D. Van Harlingen for discussions. We thank E. Bakkers, G. Badawy and S. Gazibegovic for providing InSb nanowires. We acknowledge the use of shared facilities of the NSF Materials Research Science and Engineering Center (MRSEC) at the University of California Santa Barbara (DMR 1720256) and the Nanotech UCSB Nanofabrication Facility.

**Funding information**  Work supported by the ANR-NSF PIRE:HYBRID OISE-1743717, NSF Quantum Foundry funded via the Q-AMASE-i program under award DMR-1906325, the Transatlantic Research Partnership and IRP-CNRS HYNATOQ, U.S. ONR and ARO.

**Author contributions**  A.-H.C, H.W., and M.H. grew InAs nanowires and the dielectric layer. M.P., J.S.L., C.P.D., A.P.M., S.D.H., J.T.D., and C.J.P. grew quantum wells and superconducting films. A.Z. and P.Z. fabricated devices. L.J., V.V.d.S., A.Z., and P.Z. performed measurements. L.J. and P.Z. did the simulation. A.Z. and L.J. prepared draft versions of this manuscript. P.Z. and S.M.F. wrote the manuscript with inputs from all authors.

# Supplementary material: Large second-order Josephson effect in planar superconductor-semiconductor junctions

## A  Device information

Table 1: Device information. SQUID-1 and JJ-1 are discussed in the main text. The rest are discussed in supplementary materials.

| Name | Chip name | Reference code | Name in Ref. [42] | Superconductor | Shadow wire |
|------|-----------|----------------|-------------------|----------------|-------------|
| SQUID-1 | Al-chip-3 | 20210924 Al InAs 2DEG 4.10 | SQUID-1 | Al | InAs |
| JJ-1 | Al-chip-1 | 2019 2DEG 9 | JJ-S3 | Al | InSb |
| SQUID-S1 | Al-chip-2 | 210329 Al InAs 2DEG 6.8 | - | Al | InAs |
| SQUID-S2 | Sn-chip-2 | 20211009 Sn InAs 2DEG 7.7 | - | Sn | InAs |
| SQUID-S3 | Al-chip-1 | 2019 2DEG 16 | - | Al | InSb |
| JJ-S1 | Al-chip-1 | 2019 2DEG 10 | JJ-S4 | Al | InSb |
| JJ-S2 | Al-chip-2 | 210329 Al InAs 2DEG 7.5b | JJ-S9 | Al | InAs |
| JJ-S3 | Sn-chip-2 | 20211009 Sn InAs 2DEG 9.8a | JJ-2 | Sn | InAs |

## B  Models for a single JJ

The magnetic field induces an extra phase variation inside the junction (Eq. B.3), leading to the supercurrent interference. The switching current ($I_{sw}$) of a single JJ in a magnetic field can be written as

$$I_{sw}(\Phi_{JJ}) = \max\{I(\varphi_0, \Phi_{JJ}), \quad \varphi_0 \in [0, 2\pi]\}, \tag{B.1}$$

$$I(\varphi_0, \Phi_{JJ}) = \int_0^W \frac{I_1}{W} \sin(\varphi(y)) \, dy + \int_0^W \frac{I_2}{W} \sin(2\varphi(y)) \, dy, \tag{B.2}$$

$$\varphi(y) = \varphi_0 + 2\pi\Phi_{JJ}\frac{y}{W}, \tag{B.3}$$

where $\Phi_{JJ}$ is the magnetic flux in the junction normalized by the superconducting flux quantum ($\Phi_0 = h/2e$), $\varphi_0$ is the phase difference which is a free parameter here for calculating $I_{sw}$, $W$ is the width (perpendicular to the direction of the current) of the junction, $I_1$ and $I_2$ are amplitudes of the first- and second-order harmonics in the CPR ($I_n$ can be regarded as the global current and $I_n/W$ as the current density), we ignore higher-order harmonics in our model, $y$ is the position in the $W$ direction. The integration in Eq. B.2 is independent of $W$, so we can set $W = 1$ for simplicity.

If $I_1$ is a constant and $I_2 = 0$, Eq. B.2 can be simplified as

$$I(\varphi_0, \Phi_{JJ}) = I_1 \text{sinc}(\pi\Phi_{JJ}) \sin(\varphi_0 + \pi\Phi_{JJ}), \tag{B.4}$$

and $I_{sw}$ reduces to $|I_1 \mathrm{sinc}(\pi \Phi_{JJ})|$ which resembles the Fraunhofer diffraction.

Three models for JJs are used in Fig. 4(b):

1. CPR has only the 1st harmonic, $I_1 \neq 0$, $I_2 = 0$.

2. CPR has 1st and 2nd harmonics, $I_1 \neq 0$, $I_2 \neq 0$.

3. CPR has only the 1st harmonic and $I_1$ is non-uniformly distributed. In this model, $I_1(y)$ is a three-step piece-wise function. It equals to $j_{\mathrm{center}}$ in the middle ($W/3 < y < 2W/3$) and $j_{\mathrm{side}}$ on two sides.

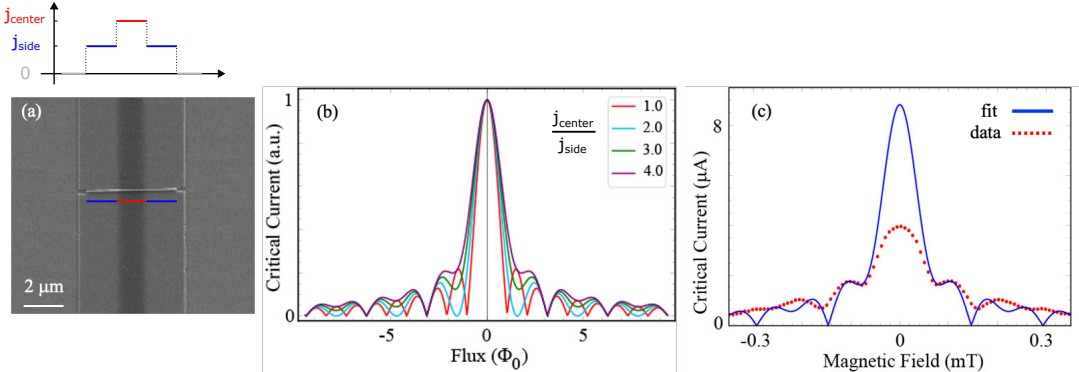

Figure 7: JJ simulation with non-uniformly distributed critical current. (a) SEM image of a device with unintentional ma-N 2403 resist residue due to a double-exposure dose. The residue regime is a dark vertical strip laying on the middle of the junction and happens to occupy roughly 1/3 of the width. We have this double-exposure issue on Al-chip-1 and get-rid of it on other chips. For more details about this fabrication issue see other sections (e.g., Fig. 29) and Ref. [42]. Here we focus on the simulation results from our model. The upper panel shows a sketch on how we model the critical current density. (b) Simulated critical current for a variety of $j_{center}/j_{side}$ values. For $j_{center}/j_{side} = 1$ the critical current resembles the Fraunhofer diffraction pattern (red). As $j_{center}/j_{side}$ increases, the first and second nodes are lifted and merge as a single dip (blue, green, and purple). This is not surprising because if $j_{side} \to 0$ ($j_{center}/j_{side} \to \infty$), we get a JJ which is 1/3 of the original width, thus 3 times in the Fraunhofer period. (c) Fit to experimental data with $j_{center}/j_{side} = 3.76$, red dots extracted from Fig. 4(a), JJ-1. This figure is adapted from Fig. 8.4 in Ref. [43]. Parameters for extracting critical current from the experimental data and for fitting are slightly different from Fig. 4.

## C   More JJ simulations

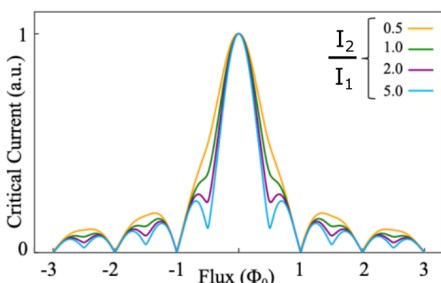

Figure 8: JJ simulation with a variety of $I_2/I_1$ values. Kinks at half periods develop as $I_2/I_1$ increases. This figure is adapted from Fig. 8.2(c) in Ref. [43].

## D   Non-inductive SQUID model

As sketched in Fig. 1(a), a SQUID consists of two JJs, i.e., JJ$_a$ and JJ$_b$. We denote magnetic fluxes (normalized by $h/2e$) in the enclosed area of the SQUID, JJ$_a$, and JJ$_b$ by $\Phi_{SQUID}$, $\Phi_{JJ,a} = \Phi_{SQUID}/r_a$ and $\Phi_{JJ,b} = \Phi_{SQUID}/r_b$. Here $r_a$ and $r_b$ are ratios between the SQUID-enclosed area and junction areas. We denote amplitudes of harmonics in JJ$_a$ and JJ$_b$ by $I_{1,a}$, $I_{2,a}$, $I_{1,b}$, $I_{2,b}$.

Similar to Eq. B.1-B.3, the switching current of a SQUID can be calculated by the following equations

$$I_{sw}(\Phi_{SQUID}) = \max\{I(\varphi_0, \Phi_{SQUID}), \quad \varphi_0 \in [0, 2\pi)\}, \tag{D.1}$$

$$I(\varphi_0, \Phi_{SQUID}) = \int_0^{W_a} \frac{I_{1,a}}{W_a} \sin(\varphi_a(y)) \, dy + \int_0^{W_a} \frac{I_{2,a}}{W_a} \sin(2\varphi_a(y)) \, dy$$
$$+ \int_0^{W_b} \frac{I_{1,b}}{W_b} \sin(\varphi_b(y) + \delta) \, dy + \int_0^{W_b} \frac{I_{2,b}}{W_b} \sin(2\varphi_b(y) + 2\delta) \, dy, \tag{D.2}$$

$$\varphi_i(y) = \varphi_0 + 2\pi \frac{\Phi_{SQUID}}{r_i} \frac{y}{W_i}, \quad i \in \{a, b\}, \tag{D.3}$$

where $\delta = 2\pi(\Phi_{JJ,a} + \Phi_{SQUID}) = 2\pi(1/r_a + 1)\Phi_{SQUID}$. $W_a$ and $W_b$ are junction widths. The integration terms are independent of widths because $W_a$ and $W_b$ are absorbed under the transformation $y \to y/W_i$.

In the simplest situation, i.e, $r_a, r_b \gg 1$, two junctions are identical, and there is only the first harmonic in the CPR, $I_{sw}$ reduces to

$$2I_{sw,JJ}(\Phi_{JJ}) |\cos(\pi\Phi_{SQUID})|, \tag{D.4}$$

which is a high-frequency SQUID oscillation (the cosine term) modulated by a low-frequency Fraunhofer oscillation ($I_{sw,JJ}$).

In our simulation, we assume $I_{2,i}/I_{1,i}$ is a constant independent of the junction index $i$ and the gate voltage. Parameters used for SQUID-1 in Fig. 3 are as follows, $r_i = 185$, $I_{1,i} = \alpha I_{sw,i}$, $I_{2,i} = 0.4\alpha I_{sw,i}$, $i \in \{a, b\}$. $r_i$ is extracted from periods of the JJ oscillation and the SQUID oscillation. $I_{sw,i}$ is the measured zero-field switching current in junction $i$ [Fig. 1(c)]. $\alpha$ is a fitting parameter which is chosen to be 0.91 in Fig. 3(a) and 0.82 in Fig. 3(b). The difference

in $\alpha$ may arise due to the uncertainty in tuning nominal switching currents by gates or higher-order harmonics that are not considered in the simulation.

# E   More SQUID simulations

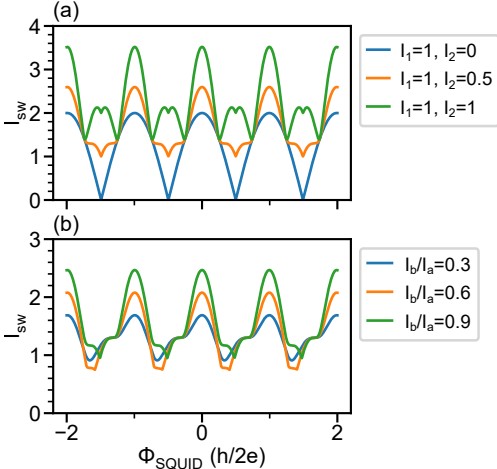

Figure 9: Simulated switching current ($I_{sw}$) as a function of SQUID flux ($\Phi_{\text{SQUID}}$). (a) In the symmetric condition, i.e., $JJ_a$ and $JJ_b$ are identical. $I_1$ and $I_2$ are amplitudes of first and second harmonics, respectively. As $I_2$ increases, $I_{sw}$ deviates from the standard $|\cos(\pi\Phi_{\text{SQUID}})|$ curve (blue) and minimums at half-integer $\Phi_{\text{SQUID}}$s are lifted from 0 (orange and green). (b) In the asymmetric condition, we fix the first ($I_{a,1}$) and the second ($I_{a,2}$) harmonics in $JJ_a$ to 1 and 0.5, respectively. Harmonics in $JJ_b$ are a fraction of those in $JJ_a$ ($I_b/I_a$). The curve becomes more symmetric as $I_b/I_a$ approaches 1, and vice versa.

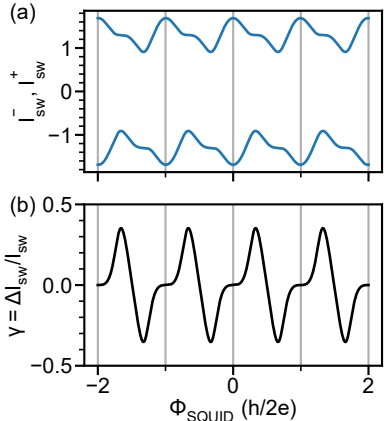

Figure 10:  Superconducting diode effect due to the second-order Josephson harmonic. (a) Duplication of Fig. 6(a). (b) Calculated $\gamma$ coefficient for the superconducting diode effect, $\Delta I_{sw} = |I_{sw}^+| - |I_{sw}^-|$, $I_{sw} = (|I_{sw}^+| + |I_{sw}^-|)/2$.

# F  Inductive SQUID

We model the inductive SQUID following Ref. [55], but including higher order terms. The currents through junctions a and b are:

$$I_a = (1-\alpha)I_0(\sin\varphi_a + \gamma\sin 2\varphi_a), \tag{F.1}$$

$$I_b = (1+\alpha)I_0(\sin\varphi_b + \gamma\sin 2\varphi_b), \tag{F.2}$$

where $\varphi_a$ and $\varphi_b$ are phase differences in junctions a and b, respectively. $\gamma$ is the amplitude of the second harmonic. For simplicity, we use normalized currents, $i_a = I_a/I_0$ and $i_b = I_b/I_0$. The total current $i$ and circulating current $j$ are:

$$i = i_a + i_b, \tag{F.3}$$

$$j = (i_b - i_a)/2. \tag{F.4}$$

$\varphi_a$ and $\varphi_b$ are connected by the equation:

$$\varphi_b = \varphi_a + 2\pi\phi_{ext} - \pi\beta j + 2n\pi, \tag{F.5}$$

where $\phi_{ext} = \Phi_{ext}/\Phi_0$ is the normalized flux applied by the external field, $\beta = 2LI_0/\Phi_0$ is the normalized inductance, $n$ is an arbitrary integer. Here we ignore the inductance difference between the two arms of the SQUID for simplicity. A more general case can be found in Ref. [55]. Note that when $\gamma \neq 0$, $I_0$ is different from the critical current by a factor $f(\gamma)$. We should use $\beta' = (I_{c,a} + I_{c,b})L/\Phi_0 = f(\gamma)\beta$ instead of $\beta$ to compare with the experimental parameters.

The maximized (minimized) $i$ is achieved when both $i_a$ and $i_b$ are maximized (minimized), which gives $\varphi_a = \varphi_b$, $i_a = \pm(1-\alpha)f(\gamma)$, $i_b = \pm(1+\alpha)f(\gamma)$, $+ (-)$ for the maximum (minimum). The external fluxes where $i$ reaches maximum (minimum) can be calculated by substituting these into Eq. F.5:

$$\phi_{ext,\pm} = \pm\alpha\beta'/2 - n. \tag{F.6}$$

The dependence of critical current on $\phi_{ext}$ is calculated using the following procedure. First, at every $\phi_{ext}$, we search for $(\varphi_a, \varphi_b)$ pairs satisfying Eq. F.5. Second, we find the minimum and maximum currents among valid $(\varphi_a, \varphi_b)$ pairs, which are the negative and positive critical currents. The results of inductive modeling can be found in Figs. 11 and 12.

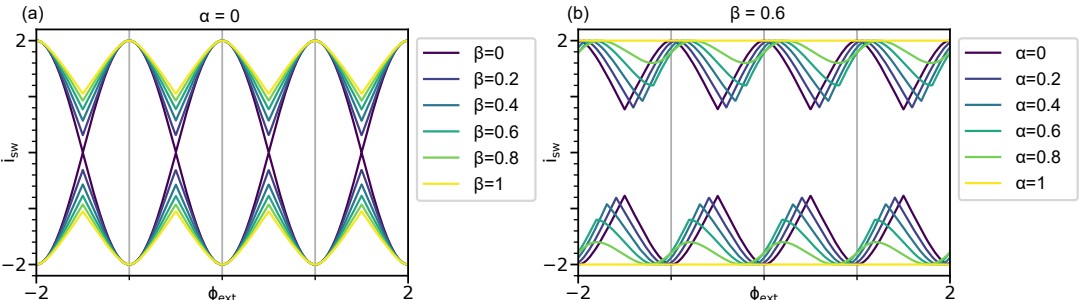

Figure 11: Simulated switching current ($i_{sw} = I_{sw}/I_0$) as a function of the normalized external flux ($\phi_{ext} = \Phi_{SQUID}/\Phi_0$), using the inductive SQUID model without the second harmonic term ($\gamma = 0$). (a) Under symmetric conditions ($\alpha = 0$). As the normalized inductance ($\beta$) increases, dips at half-integer flux values are lifted from 0. No kinks are observed near quarter flux values, which is different from results with the second harmonic (Fig. 9(a)). (b) For asymmetric conditions, we fix the normalized inductance $\beta = 0.6$ and vary $\alpha$. The ratio of critical current between two junctions is $(1 - \alpha)/(1 + \alpha)$. As the SQUID becomes more asymmetric, $i_{sw}$ becomes more skewed despite the current-phase relation being sinusoidal. No extra kinks appear (in contrast with Fig. 9(b)).

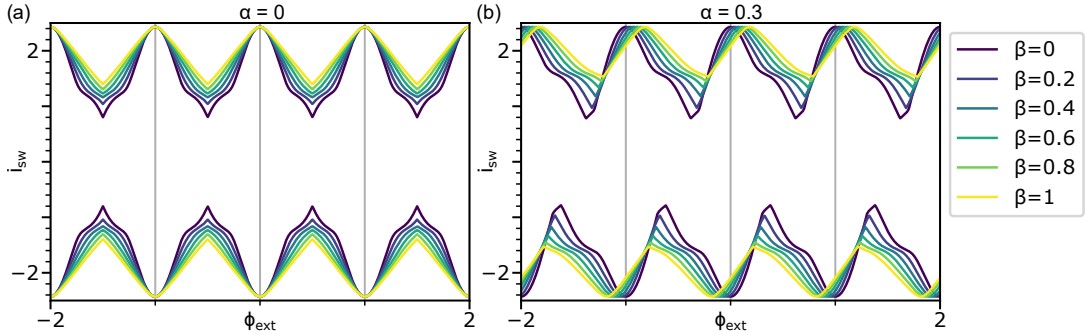

Figure 12: Simulated switching current ($i_{sw} = I_{sw}/I_0$) as a function of the normalized external flux ($\phi_{ext} = \Phi_{SQUID}/\Phi_0$), using the inductive SQUID model with the second harmonic amplitude $\gamma = -0.4$. (a) Symmetric case ($\alpha = 0$). As the normalized inductance ($\beta$) increases, tips at half-integer flux values move to larger absolute values. Kinks near quarter flux values are suppressed by increasing $\beta$. (b) Asymmetric case ($\alpha = 0.3$). As $\beta$ increases, the maxima and minima shift relative to each other horizontally, and the curve becomes more sawtooth-like. The complete information about the current-phase relation is lost at large $\beta$.

## G  Sign of the second harmonic term

In the JJ model and the non-inductive SQUID model, changing the sign of the second harmonic term is equivalent to changing the sign of the external flux. This is because Eqs. B.2 and D.2 are unchanged under the transformation $\{I_2, \Phi_{JJ}, \varphi_0\} \rightarrow \{-I_2, -\Phi_{JJ}, \pi - \varphi_0\}$ and $\{I_{2,a}, I_{2,b}, \Phi_{SQUID}, \varphi_0\} \rightarrow \{-I_{2,a}, -I_{2,b}, -\Phi_{SQUID}, \pi - \varphi_0\}$, respectively.

In the inductive SQUID model, changing the sign of the second harmonic term is equivalent to changing both signs of the external flux and the normalized inductance $\beta$. This is because Eqs. F.1 and F.2 are unchanged under the transformation $\{\gamma, \phi_{ext}, \beta, \varphi_a\} \rightarrow \{-\gamma, -\phi_{ext}, -\beta, \pi - \varphi_a\}$.

The results for flipping the signs of coefficients in above models can be found in Figs. 13 and 14.

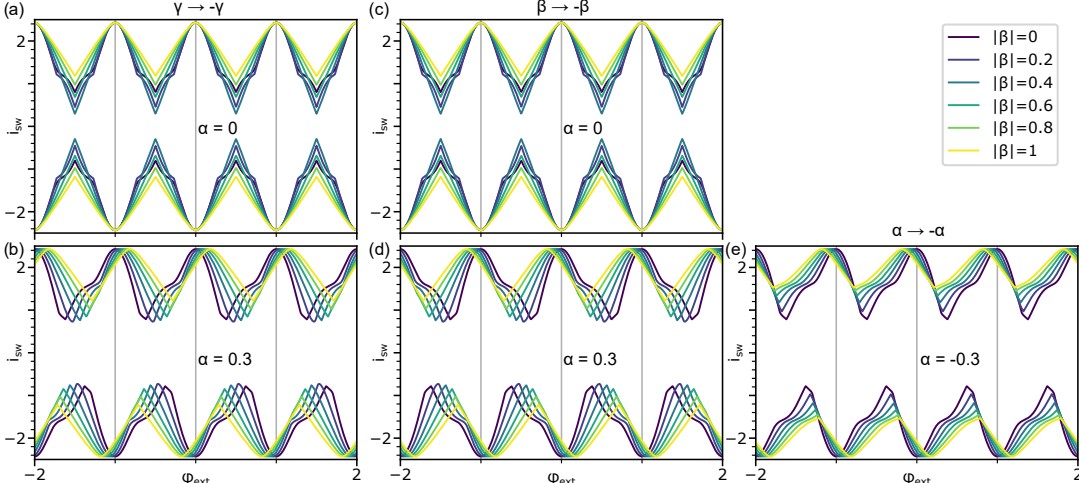

Figure 13: Changing the signs of parameters in the inductive SQUID model, to be compared with Fig. 12. (a)(b), (c)(d), and (e) changing the signs of the second harmonic term $\gamma$, the normalized inductance $\beta$, and the junction asymmetry $\alpha$. We observe that changing the sign of $\gamma$ is equivalent to changing both the sign of $\beta$ and $\varphi_{ext}$, while changing the sign of $\alpha$ is equivalent to changing the sign of the external flux (flipping the figure along the y axis). Note that $\beta$ is positive in these devices. The sign of $\gamma$ can be determined by the evolution of $I_{sw}$ dips (maxima) against $|\beta|$ in the symmetric (asymmetric) case.

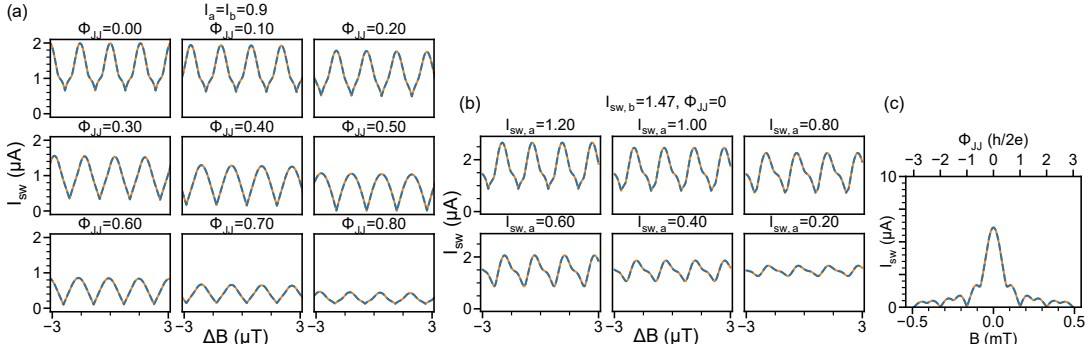

Figure 14: Changing the signs of the second harmonic term and the external flux simultaneously keeps the result unchanged in the model without inductance. Orange curves are the same simulations as orange curves in Figs. 3 and 4(b), while dashed blue curves are simulations with opposite signs of both the second harmonic term and the external flux.

# H Supplementary data from SQUID-1

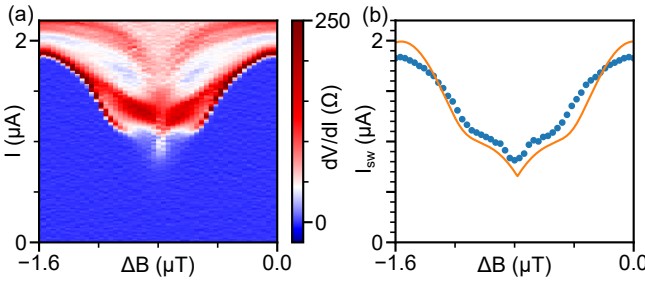

Figure 15: (a), (b) Zoomed-in data of Figs. 2(a) and 3(a) ($\Phi_{JJ} = 0$), respectively.

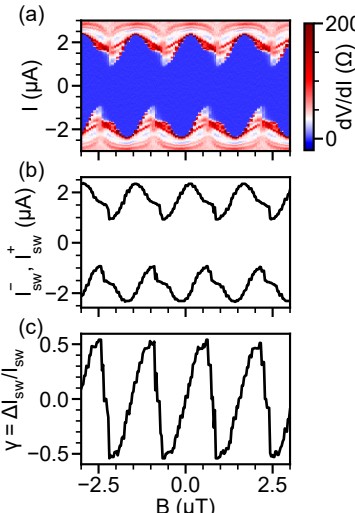

Figure 16: Non-identical negative and positive critical currents (superconducting diode effect) in SQUID-1 when tuned to the asymmetric regime. $V_{g,a} = 100$ mV, $V_{g,b} = 500$ mV. (a) Duplication of Fig. 6(b). (b) Extracted negative and positive switching currents from panel (a). (c) Calculated $\gamma$ coefficient for the superconducting diode effect, $\Delta I_{sw} = |I_{sw}^+| - |I_{sw}^-|$, $I_{sw} = (|I_{sw}^+| + |I_{sw}^-|)/2$.

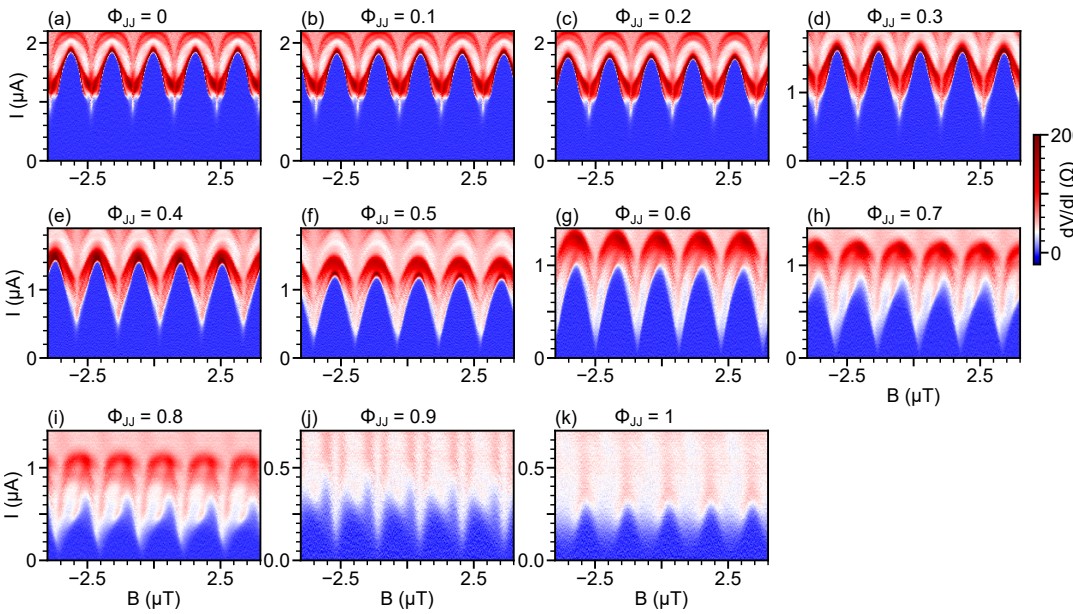

Figure 17: Supplementary data to Fig. 3(a). SQUID oscillation in SQUID-1 at a variety of $\Phi_{JJ}$s which are noted at the top of each panel. $V_{g,a} = 128$ mV, $V_{g,b} = 113$ mV. $I_{sw,a} = I_{sw,b} = 0.9$ $\mu$A.

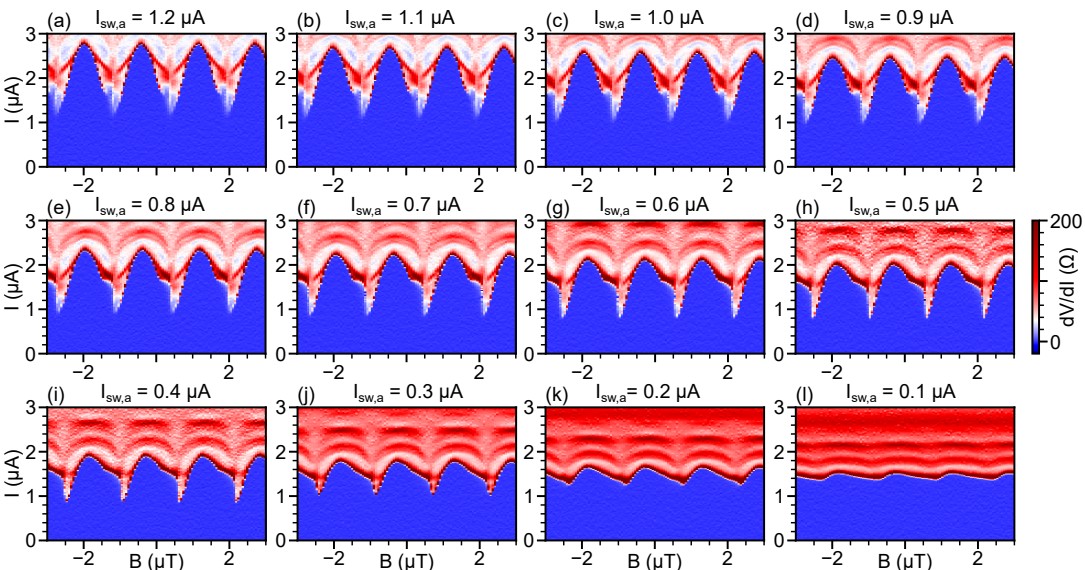

Figure 18: Supplementary data to Fig. 3(b). SQUID oscillation in SQUID-1 at different $I_{sw,a}$s which are noted at the top of each panel. $I_{sw,b} > I_{sw,a}$ is fixed at 1.47 $\mu$A. $\Phi_{JJ} = 0$.

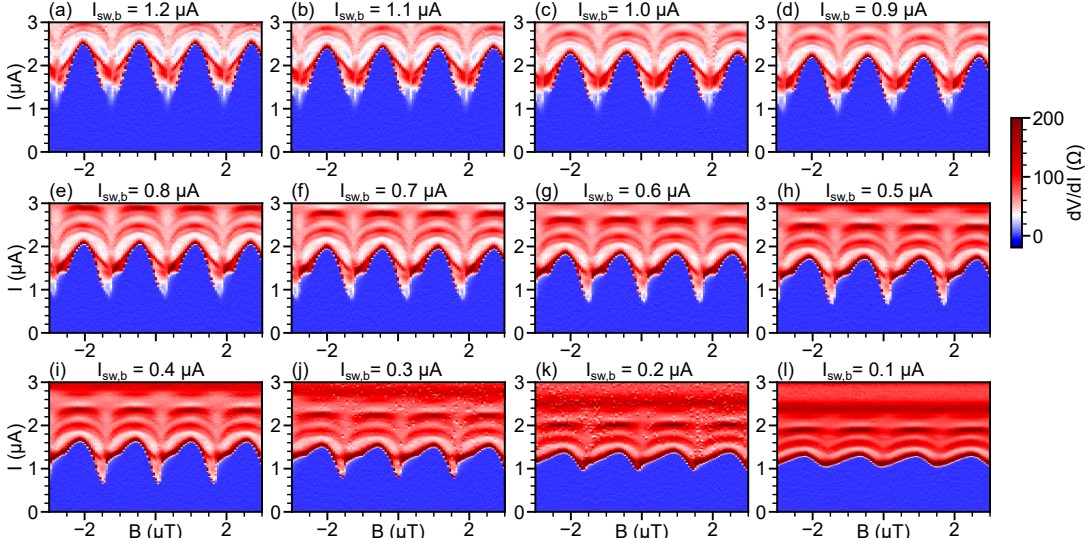

Figure 19: SQUID oscillation in SQUID-1 at a variety of $I_{sw,b}$s which are noted at the top of each panel. $I_{sw,a} > I_{sw,b}$ is fixed at 1.22 $\mu$A. $\Phi_{JJ} = 0$. The half-periodic kink position and the skew direction flip comparing to those in Fig. 18.

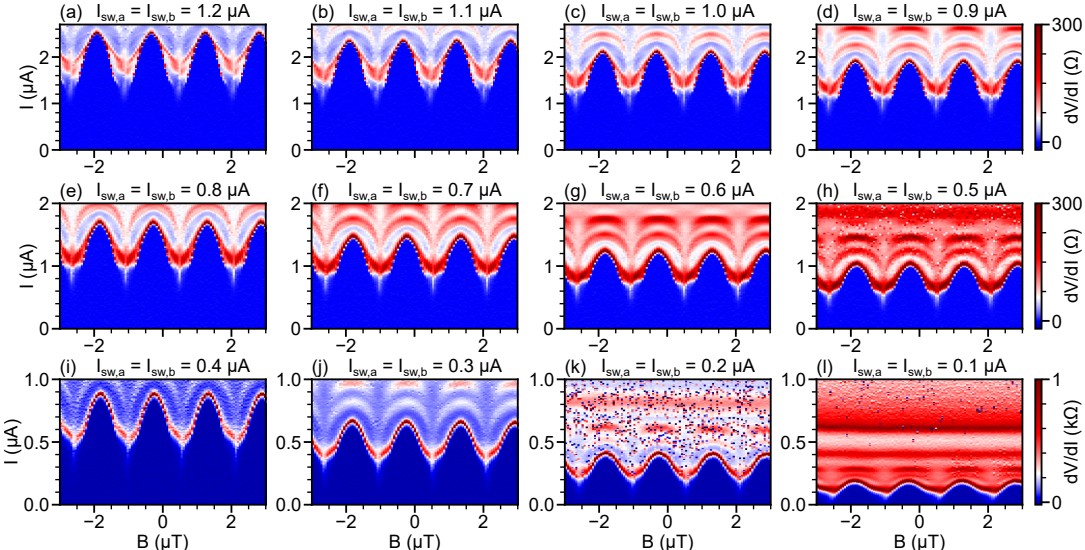

Figure 20: SQUID oscillation in SQUID-1 when $I_{sw,a} = I_{sw,b}$ which are noted at the top of each panel. $\Phi_{JJ} = 0$.

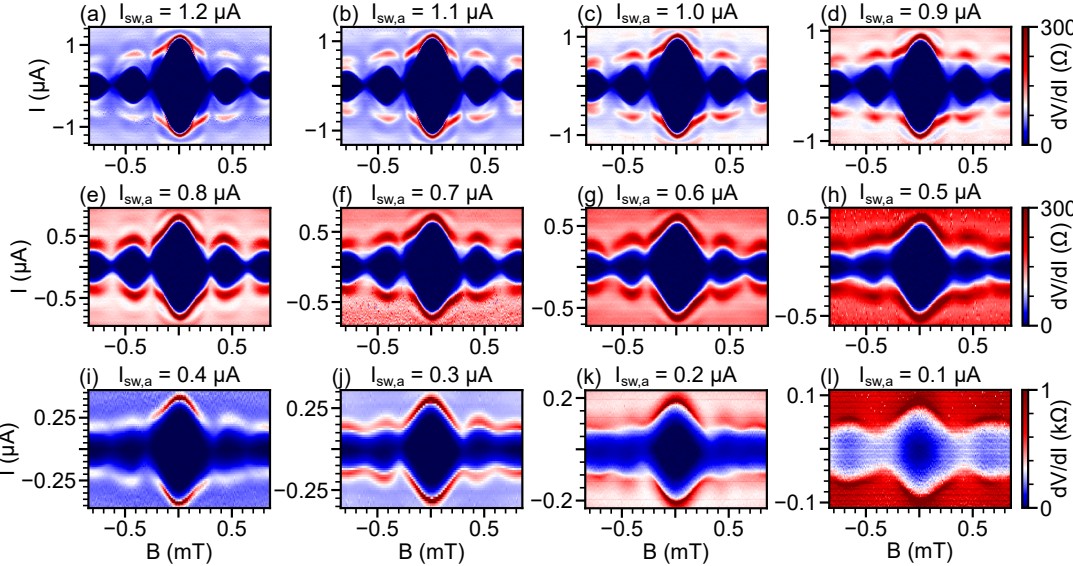

Figure 21: Single JJ oscillation in SQUID-1 at a variety of $I_{sw,a}$s which are noted at the top of each panel. $I_{sw,b}$ is tuned to 0.

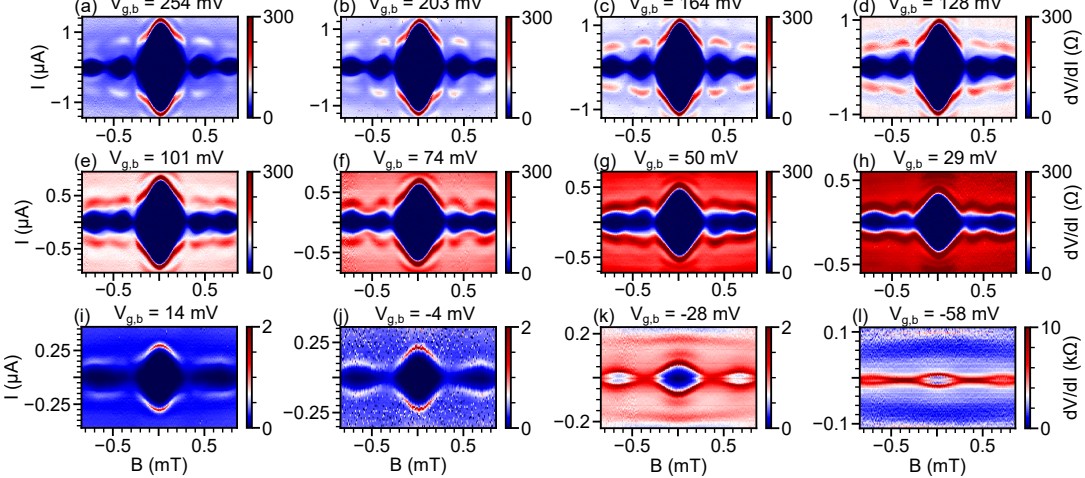

Figure 22: Single JJ oscillation in SQUID-1 at a variety of $V_{g,b}$s which are noted at the top of each panel. $I_{sw,a}$ is tuned to 0. Note that here we show $V_{g,b}$ instead of $I_{sw,b}$.

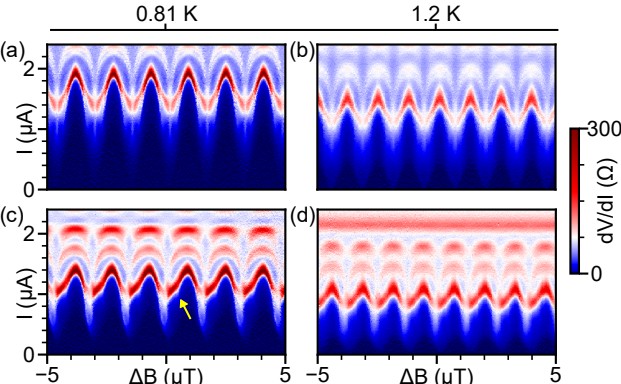

**Figure 23:** Switching current ($I_{sw}$) oscillation at higher temperatures in SQUID-1. The temperature is noted at the top axis. (a-b) The SQUID is tuned to a symmetric state. Nominal zero-field switching currents in two junctions are $I_{sw,a} = I_{sw,b} = 1\,\mu$A. (c-d) The SQUID is tuned to an asymmetric state. $I_{sw,a} = 1\,\mu$A, $I_{sw,b} = 0.42\,\mu$A. Half-periodic kinks are still clear at 0.81 K (yellow arrow).

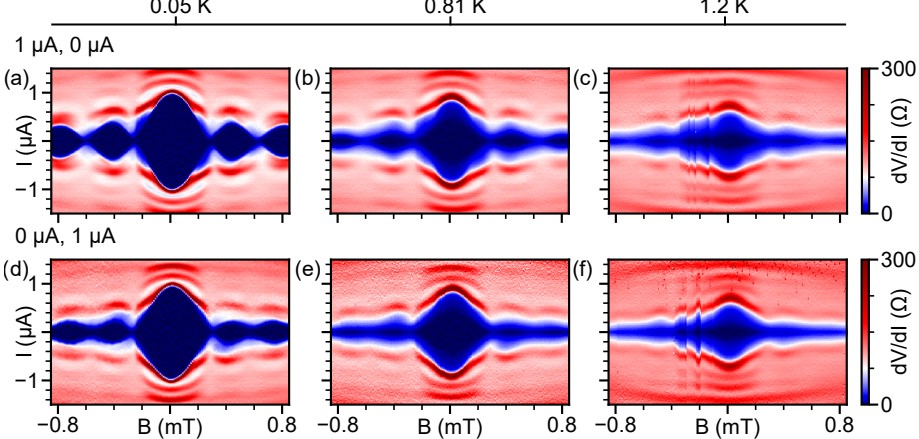

**Figure 24:** Temperature dependence of single JJ diffraction patterns. The temperature is noted at the top axis. (a-c) $I_{sw,a} = 1\,\mu$A, $I_{sw,b} = 0$. (d-f) $I_{sw,a} = 0$, $I_{sw,b} = 1\,\mu$A.

# I   Data from SQUIDs not shown in the main text

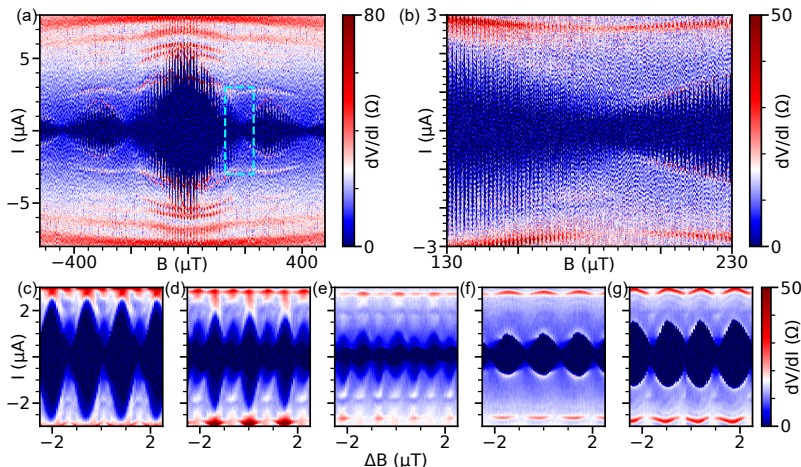

Figure 25:   Data from SQUID-S1 which is similar to SQUID-1 except that mask nanowires are not connected to electrodes. (a) Measured differential resistance as a function of current bias and magnetic field threading the SQUID. (b) Zoomed-in data of the regime enclosed by the dashed rectangle in panel (a) . (c-g) Zoomed-in data of panel (b) taken near $B_0$ equals to 130, 155, 180, 205, and 230 $\mu$T, respectively. $\Delta B$ is the deviation of the magnetic field from $B_0$. Similar to SQUID-1, the half-periodic oscillation is more obvious in panels (d) and (e) when $\Phi_{jj}$ is approaching 1.

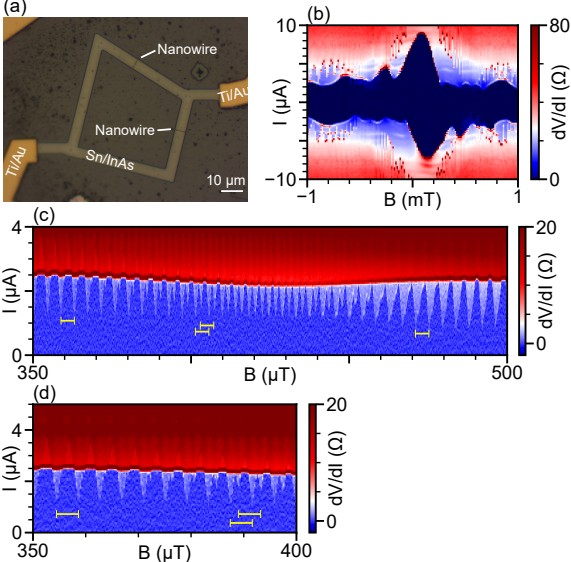

Figure 26:   Data from SQUID-S2 which is made of Sn/InAs 2DEG. (a) Optical microscope image. (b) Differential resistance as a function of the current and the field. Two superconducting transitions, instead of one, are observed. The smaller switching current does not have a high frequency SQUID component. This extra transition may be due to breaks in the superconducting film outside the SQUID loop [42]. (c) Zoomed-in data of the SQUID oscillation shows half-periodic oscillation near 400 $\mu$T. Yellow scale bars are indicators for the fundamental period and have the same length. (d) Zoomed-in data of panel (c).

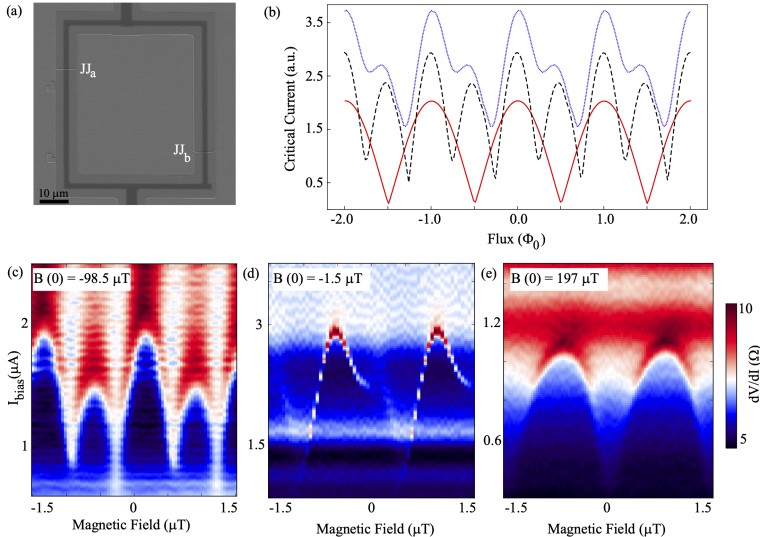

Figure 27: SQUID-S3. (a) SEM of the device. The dark strip is the residue of the e-beam resist (ma-N 2403) due to double exposure. (b) Simulations of SQUID characteristics with two-component CPR that resemble data in this panel. (c-e) SQUID oscillation at different background fields which are noted in the top-left of each panel. Switching current pattern has double the frequency in panel (c) compare to panel (e) going through a transition regime in panel (d). A vertical line-cut is subtracted in panels (c) and (e) to remove a spurious superconducting switching transition from an uncontrolled junction elsewhere in the circuit (see Fig. 28).

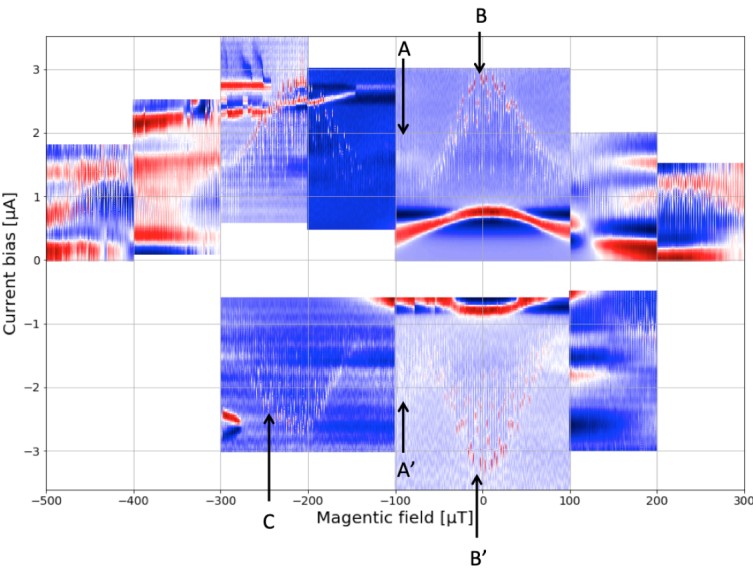

Figure 28: SQUID oscillation in SQUID-S3 on a vast span of field. The device shows multiple superconducting transition as well as a flux difference between two junctions. Fluxes in the two junctions of the SQUID were offset, presumably due to trapped flux, which resulted in the Fraunhofer pattern maxima at different values of global flux. This allowed to explore SQUIDs with different ratios of $I_1$ and $I_2$ as a function of a single global junction flux control knob, without gates that were not yet developed at that time.

## J Supplementary data from JJs not shown in the main text

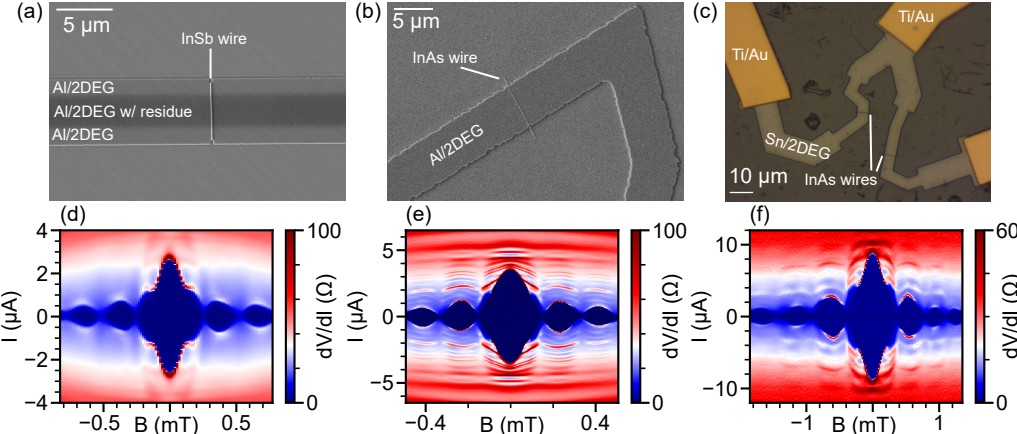

Figure 29: Half-periodic kinks from three different types of JJs made with the nanowire shadow method. (a) SEM image of the first type of JJs which is made from Al/InAs 2DEG and bared InSb shadowing wires (Al-chip-1). The InSb nanowire is etched during the etching for making the Al/2DEG mesa. The dark horizontal strip is unintentional ma-N 2403 resist residue due to a double-exposure dose. JJ-1 and JJ-S1 belong to this type (including the residue). (b) SEM image of the second type of JJ which is also made from Al/InAs 2DEG but the shadow nanowire is InAs with a $HfO_x$ capping layer (Al-chip-2 and Al-chip-3). The $HfO_x$ layer protects the nanowire from being etched. Wires can be contacted by Ti/Au electrodes to work as self-aligned gates like those in SQUID-1 [Fig. 1(b)]. (c) Optical microscope image of the third type of JJ which is similar to the second type except that Al is replaced by Sn and leads are partially covered by Ti/Au to short possible unintentional breaks on the leads (Sn-chip-2). (d-f) Superconducting diffraction patterns from devices JJ-S1, JJ-S2, and JJ-S3 which are similar to or are the devices in (a-c), respectively. All three kinds of devices show kinks at half-periods. Data from more JJs are available in the supplementary materials of Ref. [42].

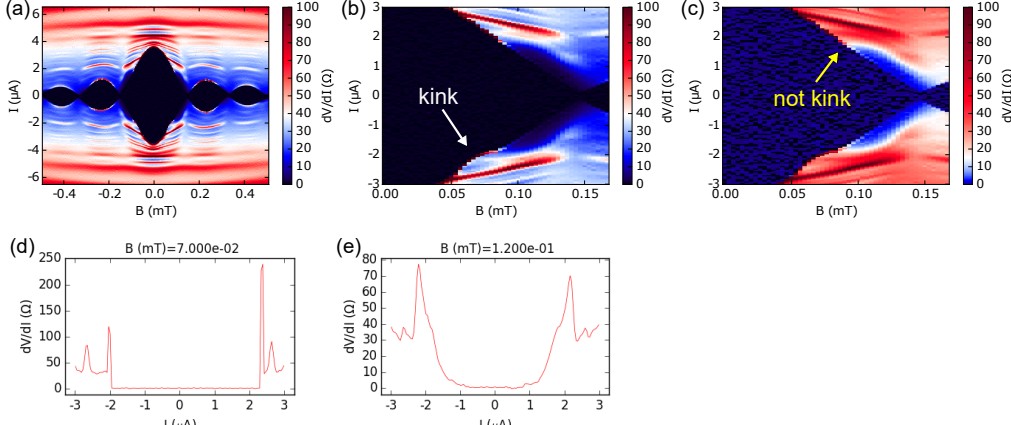

Figure 30: Details near a kink. Data from JJ-S2. (a) Replotted from the same dataset as Fig. 29(e). (b) Zoomed-in data of panel (a). The white arrow points at an apparent kink in negative bias where the superconducting-normal state switching boundary is sharp and the slope of the switching current in applied flux changes visibly. (c) Same as panel (b) except the colors are adjusted. We observe that the boundary of the true zero resistivity region is straight and exhibits no kink. However, there is an additional blue bump just outside the boundary which is a region of low resistivity that is flux-dependent. Because of the low value of resistivity, depending on the exact method we use to extract the switching current there maybe a kink in the extracted curve. (d),(e) Linecuts at fixed magnetic fields indicated in titles.

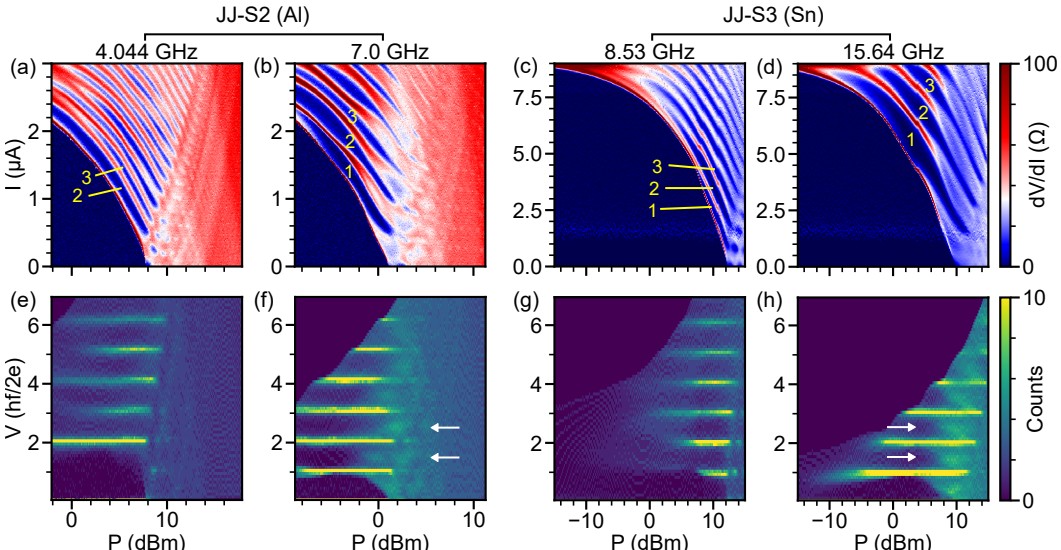

Figure 31: Shapiro steps from JJ-S2 (Al, left panels) and JJ-S3 (Sn, right panels) at zero field. The microwave frequency is noted at the top. (a-d) Differential resistance as a function of the current and the microwave power. Shapiro steps manifest as dips in $dV/dI$, some of which are indicated by yellow numbers. Peaks between Shapiro steps split at higher frequencies. (e-h) Histogram of the voltage as a function of the microwave power. Shapiro steps are peaks in histograms. Half-integer steps at high frequencies are highlighted by white arrows in panels (f) and (h). JJ-S2 has a missing first step at 4.044 GHz which is likely due to self-heating. More discussion about the missing Shapiro steps are available in Ref. [49].

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
