# Peer review of "Large Second-Order Josephson Effect in Planar Superconductor-Semiconductor Junctions"

_SciPost Physics, doi:SciPost Phys. 16, 030 (2024)_

## Round 2 · Referee Report · Anonymous (Referee 2) · 2023-5-30

Strengths

skillful work in the field of mesoscopic physics, provides important advances in studies of the Josephson effect

Weaknesses

Related previous research is not fully cited

Report

This is interesting and important contribution to the field of mesoscopic physics. The manuscript reports the results of investigation of the current-phase relations (CPR) of Al/InAs-quantum well planar Josephson junctions fabricated using nanowire shadowing technique. The junctions exhibit large second-order CPR harmonic. The results will stimulate further theoretical work on explanation of the observed effects. This work should be of interest to broad community of researchers working in the field of superconducting hybrid structures.

I can recommend the manuscript for publication in SciPost provided the authors will complete the citation list

Requested changes

  1. The authors correctly state that " In the clean SNS limit, the CPR is predicted to be linear or a skewed sine function [7–10]"

In this respect, one should cite original paper I.O. Kulik and A. N. Omelyanchuk, JETP Lett. 21, 96 (1975)

  1. Regarding second-harmonic CPR high-temperature superconductor junctions, relevant paper is

E. Il’ichev et al., Degenerate Ground State in a Mesoscopic YBa2Cu3O72x Grain Boundary Josephson Junction, PRL 86, 5369 (2001)

  1. The manuscript will benefit from citation of other relevant studies of 2nd harmonic in CPR

L. V. Ginzburg, I. E. Batov, V. V. Bol'ginov, S. V. Egorov, V. I. Chichkov, A. E. Shchegolev, N. V. Klenov, I. I. Soloviev, S. V. Bakurskiy, and M. Yu. Kupriyanov JETP Lett. 107, 48 (2018)

M. Kayyalha, M. Kargarian, A. Kazakov, I. Miotkowski, V. M. Galitski, V. M. Yakovenko, L. P. Rokhinson, and Y. P. Chen, Phys. Rev. Lett. 122, 047003 (2019)

F. Nichele, E. Portolés, A. Fornieri, A. M. Whiticar, A. C. C. Drachmann, S. Gronin, T. Wang, G. C. Gardner, C. Thomas, A. T. Hatke, M. J. Manfra, and C. M. Marcus, Phys. Rev. Lett. 124, 226801 (2020)

M. Endres, A. Kononov, H. S. Arachchige, J. Yan, D. Mandrus, K. Watanabe, T. Taniguchi, C. Schönenberger, arXiv.2211.10273 (2022)

I. Babich, A. Kudriashov, D. Baranov, V. Stolyarov, arXiv:2302.02705 (2023)

C. Li, J. C. de Boer, B. de Ronde, S. V. Ramankutty, E. van Heumen, Y. Huang, A. de Visser, A. A. Golubov, M. S. Golden, and A. Brinkman, Nat. Mater. \textbf{17}, 875 (2018)

  • validity: high
  • significance: high
  • originality: high
  • clarity: high
  • formatting: perfect
  • grammar: perfect

Author:  Po Zhang  on 2023-11-21  [id 4134]

(in reply to Report 2 on 2023-05-30)

We thank the referee and added Refs. We did not include Nat. Mater. \textbf{17}, 875 (2018). This work, accompanied by several other papers of similar impact, is about missing Shapiro steps and the sin(phi/2) term. We briefly mentioned the missing Shapiro steps in the third paragraph on page 5 and refer readers to a detailed discussion in a parallel manuscript (Ref. 50, by the same main authors).

---

## Round 2 · Referee Report · Anonymous (Referee 1) · 2023-5-30

Strengths

1-Clarity
2-Conciseness

Report

The work reports measurements on planar Josephson junctions, focused on the study of the presence of higher harmonics in the CPR. The main results is the assessment that the second-order harmonic, the sin(2phi) term, is large. The assessment is based on three different measurements: SQUID measurements, single-junction diffraction patterns, and Shapiro steps, and so it is quite comprehensive. It is critically discussed and some remaining mysteries are

The topic of this work is quite timely, since the non-linearity of these higher harmonics can have interesting applications in e.g. quantum devices or superconducting diodes. This work has the merit of critically discussing the origin of the effect, and to discuss some of the pitfalls that may occur when interpreting such measurements, also on the related topic of the fractional Josephson effect, the sin(phi/2) term. Furthermore, it is clear and concise.

I find that this paper essentially meets the general acceptance criteria, and that it also provides a pathway for studying higher harmonics in planar Josephson junctions, a line of research that I expect to allow quite a lot of future explorations. I recommend publication, while also providing a list of minor clarifications requests below.

Requested changes

I have the following questions that may lead to minor revisions and improvements of the work:

1) The abstract states that the junctions exhibit an "unusally large second-rder Josephson harmonic". What is the point of comparison? I do not have prior knowledge or quantitative intuition to immediately agree that the estimated ratio I_1/I_2=0.4 is "unusually large". Somewhere in the text, it would be good to quantify this qualifier, either by referring to previous measurements, or, if unavailable, to theoretical expectations.

2) In the "Background: Current Phase Relations" part, the Authors write that "the CPR of a SIS Josephson junction is sinusoidal". Yet, higher-order terms have been observed even in such junctions (arXiv:2302.09192). The effect is smaller than what reported here of course, but it may deserve a mention as evidence that higher harmonics are ubiquitous.

3) As higher harmonics are generally present in semiconductor-based Josephson junctions, the references listed in "Previous work" seems a bit narrow. For instance, higher harmonics have been measured in nanowires also away from the 0-pi transition of a quantum dot - see e.g. PRL 115, 127002 (2015) and PRL 125, 056801 (2020). There may be other examples as well where the strength of the second harmonic was explicitly extracted.

4) In the "List of Results", the authors write that "The large sin(2phi) term is not related to a cancellation of this first term". I do not understand this sentence. To my understanding the presence of a second harmonic in the CPR is not causally related to the cancellation of the first harmonic. It's just that suppressing the first harmonic may make it easier to observe the second harmonic and estimate its magnitude (as the authors also write one paragraph above: "In both cases the second harmonic can be observed because the first harmonic is canceled"). It is not clear to me whether the Authors are making a distinction between their measurements and preivous measurements, and what this distinction is.

5) When discussing Figure 1, the Authors write that "we get an effective junction [...] an order of magnitude larger than the typical physial length". I take this to mean, as explained later in the paper, that the length extracted from the Fraunhofer period does not match at all the litographic distance between the superconductors. This fact is surprising and very interesting. While it is not the main topic of the work, I wished the Authors dwelled on it a bit more. What are possible explanations? Does the discrepancy tell us something about the nature of the proximitized 2DEG states?

6) On general grounds one expects that the ratio I_1/I_2 will depend on the gate voltage, which controls the average channel transparency in the wide junction (see e.g. the data in Fig. 3b). Is the quoted value I_1/I_2=0.4 specific for some gate voltages?

7) The article presents the missing observation of even higher harmonics as a puzzle to be solved by future work. What was the sensitivity of these measurements to higher harmonics in the CPR? In other words, how large should these terms have been in other to be clearly observed in these measurements? I imagine that the magnitude of the harmonics decreases sharply unless the CPR is very skewed; ratios I_3/I_1 could easily be around 1% even for transparent junctions.

  • validity: good
  • significance: good
  • originality: ok
  • clarity: high
  • formatting: good
  • grammar: reasonable

Author:  Po Zhang  on 2023-11-21  [id 4135]

(in reply to Report 1 on 2023-05-30)

1) The abstract states that the junctions exhibit an "unusally large second-rder Josephson harmonic". What is the point of comparison? I do not have prior knowledge or quantitative intuition to immediately agree that the estimated ratio I_1/I_2=0.4 is "unusually large". Somewhere in the text, it would be good to quantify this qualifier, either by referring to previous measurements, or, if unavailable, to theoretical expectations.

To narrowly answer the referee’s question: the point of comparison is to the first harmonic, and to the third. It is most straightforward to conclude that the second-order harmonic is unusually large based on qualitative observations: the switching current of JJs and SQUIDs shows double-oscillations, which have been observed previously under exotic circumstances such as near a 0-pi junction transition point where the first harmonic is dramatically suppressed. Technically, this can also mean that the third-order term is unusually small.

We modified the conclusions section to add a paragraph with a discussion of this.

2) In the "Background: Current Phase Relations" part, the Authors write that "the CPR of a SIS Josephson junction is sinusoidal". Yet, higher-order terms have been observed even in such junctions (arXiv:2302.09192). The effect is smaller than what reported here of course, but it may deserve a mention as evidence that higher harmonics are ubiquitous.

The originally derived CPR for an ideal SIS junction is sinusoidal. In real SIS devices, conducting channels may exist and the CPR could deviate from the standard sinusoidal curve. We rephrased the description and added arXiv:2302.09192 as a reference (Ref. 7).

3) As higher harmonics are generally present in semiconductor-based Josephson junctions, the references listed in "Previous work" seems a bit narrow. For instance, higher harmonics have been measured in nanowires also away from the 0-pi transition of a quantum dot - see e.g. PRL 115, 127002 (2015) and PRL 125, 056801 (2020). There may be other examples as well where the strength of the second harmonic was explicitly extracted.

Indeed there were works recently where second-order effects manifest or are brought out in various junctions. Too many to list them all, but the readers could indeed benefit from a few more references. Sin2phi is also the foundation for the “0-pi qubit” on which there has been some work as of late. These are clearly relevant to our work and we have now added sentences discussing the connection.

Modifications: In the “Previous Work” section, we added a new paragraph discussing experiments about SQUIDs consisting of high transparency SNS junctions, biased to half-flux quantum. We also added references for 0-pi transition in S-dot-S systems (Ref. 23-24).

4) In the "List of Results", the authors write that "The large sin(2phi) term is not related to a cancellation of this first term". I do not understand this sentence. To my understanding the presence of a second harmonic in the CPR is not causally related to the cancellation of the first harmonic. It's just that suppressing the first harmonic may make it easier to observe the second harmonic and estimate its magnitude (as the authors also write one paragraph above: "In both cases the second harmonic can be observed because the first harmonic is canceled"). It is not clear to me whether the Authors are making a distinction between their measurements and preivous measurements, and what this distinction is.

It was an awkward phrasing. It is not the second term itself but its dramatic manifestation that is not related to the cancellation of the first term.

5) When discussing Figure 1, the Authors write that "we get an effective junction [...] an order of magnitude larger than the typical physial length". I take this to mean, as explained later in the paper, that the length extracted from the Fraunhofer period does not match at all the litographic distance between the superconductors. This fact is surprising and very interesting. While it is not the main topic of the work, I wished the Authors dwelled on it a bit more. What are possible explanations? Does the discrepancy tell us something about the nature of the proximitized 2DEG states?

This is actually very standard and related to the London penetration depth. Not related to new physics.

6) On general grounds one expects that the ratio I_1/I_2 will depend on the gate voltage, which controls the average channel transparency in the wide junction (see e.g. the data in Fig. 3b). Is the quoted value I_1/I_2=0.4 specific for some gate voltages?

It appears to not be significantly gate voltage dependent, though adding a gate to these junctions was motivated by looking into this question. We fixed I_1/I_2=0.4 in both junctions of the SQUID, for all gate voltages and external fields in Fig. 3. This model, though simple, works well in this parameter range.

7) The article presents the missing observation of even higher harmonics as a puzzle to be solved by future work. What was the sensitivity of these measurements to higher harmonics in the CPR? In other words, how large should these terms have been in other to be clearly observed in these measurements? I imagine that the magnitude of the harmonics decreases sharply unless the CPR is very skewed; ratios I_3/I_1 could easily be around 1% even for transparent junctions.

In transparent and ballistic junctions, where multiple harmonics are present, the periodicity of SQUID patterns and other DC Josephson characteristics does not exhibit a second modulation, only some skew. In particular the fraction of the third harmonic should be about 25%. But also the fourth, firth etc are still significant. However, observing a doubly modulated set of characteristics is unusual, and it implies that the higher harmonics are suppressed dramatically compared to what is expected for ballistic junctions. We now comment on this in the Conclusions block of the paper.

Anonymous on 2023-11-21  [id 4138]

(in reply to Po Zhang on 2023-11-21 [id 4135])

I thank the Authors for their clarifications and modifications to the manuscript. They are all clear to me, except the reply to point (5).

"This is actually very standard and related to the London penetration depth. Not related to new physics."

To say that something is "very standard" is, byt itself, no explanation. I think the Authors should expand on this, if needed through providing references; if they have a very standard explanation of their remark, it should be added to the text.

Other than this point, I think the paper is ready for publication.

(Note: I am the referee of Report 1).

---

## Round 2 · Referee Report · Anonymous (Referee 3) · 2023-6-9

Strengths

  1. The experimental data very clearly support the conclusion that a large second harmonic is present in the current-phase relation of InAs quantum well Josephson junctions. This is seen in the modulation of the critical current of a SQUID by an applied magnetic field, as well as by observing half-integer Shapiro steps.

  2. The observation of higher harmonics is important in determining the current-phase relation of this particular type of Josephson junction (quantum wells with long elastic mean free path in combination with strong spin-orbit coupling).

  3. The experimental observations can be fitted with a simple and intuitive model in which only a second harmonic is assumed.

Weaknesses

  1. In the final paragraph before the conclusion the authors address the difference in switching current at positive and negative bias. Usually, in a SQUID this behavior can be understood from inductive effects. But the authors also refer to the Josephson diode effect in which a combination of broken symmetries are needed. Do the authors imply/suggest that this is the case here too? Have they tried to fit the usual behavior (estimate the inductance of the self-field)?

  2. Some minor issues are listed below

Report

After a quick clarification of what the authors want to say about the asymmetry in the switching currents and after addressing the minor comments I would be happy to recommend to publish this work.

Requested changes

  1. The kinks in Figs. 2 and 3 are hard to discern. Can the authors enlarge some panels of the figures to make the kinks more visible?

  2. When fitting the Fraunhofer oscillation period, the authors mention that the effective junction length is an order of magnitude larger than the physical length. When taking flux focusing by the leads into account (one triangle in every lead), the actual effective junction area is already a lot closer to what is measured/expected. Would it not make more sense to compare to this value? Otherwise the reader might get the impression that novel physics is at play, which should not be implied here.

  3. The authors say that the skewed current-phase relation of a short and ballistic junction should have higher harmonics, which are not seen in the experiment. For a Kulik-Omel'yancuk junction, what amplitude would the sin(3\phi) component have with respect to the sin(2\phi)? If this is very small (which I think it is), it might be present in the data but burried in the noise. In that case, the claim cannot be made that the reason for the sin(2\phi) lies in physics beyond the usual skewed CPR.

  4. Some experiments indicate a large cos(2\phi) contribution to the CPR. Can the authors rule out that their second harmonic is of cos(2\phi) shape rather than sin(2\phi)? If so, then it would be nice to mention

  • validity: high
  • significance: high
  • originality: high
  • clarity: good
  • formatting: excellent
  • grammar: excellent

Author:  Po Zhang  on 2023-11-21  [id 4133]

(in reply to Report 3 on 2023-06-09)

1. In the final paragraph before the conclusion the authors address the difference in switching current at positive and negative bias. Usually, in a SQUID this behavior can be understood from inductive effects. But the authors also refer to the Josephson diode effect in which a combination of broken symmetries are needed. Do the authors imply/suggest that this is the case here too? Have they tried to fit the usual behavior (estimate the inductance of the self-field)?

The asymmetry due to sin2phi results in the altered order of kink and maximum for positive vs negative bias while the asymmetry due to loop inductance results in shifted maxima for positive vs negative bias.

We thank the referee for making us think more deeply about the inductive effect. Simulations including both the second-harmonic effect and the inductive effect not only help us answer this question but also lead to a new observation that the sign of the second-harmonic term can be inferred with the inductive effect, and it happens to be negative. We added Fig 6 and a discussion. Simulations with both inductive and second harmonic effects are added as new sections in the supplementary (sections VII and VIII).

1. The kinks in Figs. 2 and 3 are hard to discern. Can the authors enlarge some panels of the figures to make the kinks more visible?

We added Fig. S9 which contains enlarged panels for Figs. 2(a) and 3(a) (\Phi_{JJ} = 0) and noted it in the main text.

We have also shared all raw data, figure data, and data-processing code on Zenodo (Ref. 58) so readers can do further analysis themselves.

Please note that there are a lot of examples of kinked SQUID curves in the supplementary and in extended data on Zenodo. We hope that the readers will look at all the data. We chose data for Figures 2 and 3 because it is a complete set that can be fit by one family of curves within the simple model.

2. When fitting the Fraunhofer oscillation period, the authors mention that the effective junction length is an order of magnitude larger than the physical length. When taking flux focusing by the leads into account (one triangle in every lead), the actual effective junction area is already a lot closer to what is measured/expected. Would it not make more sense to compare to this value? Otherwise the reader might get the impression that novel physics is at play, which should not be implied here.

We are not attempting to claim any new physics from this and we clarified this in the text. The heuristic proposed by the referee may indeed be helpful in some situations, but not generally.

3. The authors say that the skewed current-phase relation of a short and ballistic junction should have higher harmonics, which are not seen in the experiment. For a Kulik-Omel'yancuk junction, what amplitude would the sin(3\phi) component have with respect to the sin(2\phi)? If this is very small (which I think it is), it might be present in the data but burried in the noise. In that case, the claim cannot be made that the reason for the sin(2\phi) lies in physics beyond the usual skewed CPR.

The amplitude of the third-order harmonic is 64% of that of the second-order harmonic when transparency is 1 (see our replies to the referee in report 1), this is not small.

Even if this value is small, in a Kulik-Omel'yancuk junction the CPR never shows double-oscillations.

4. Some experiments indicate a large cos(2\phi) contribution to the CPR. Can the authors rule out that their second harmonic is of cos(2\phi) shape rather than sin(2\phi)? If so, then it would be nice to mention.

We do not expect these to be cosine terms. Cosine terms exist only in special junctions such as superconductor-ferromagnet-superconductor junctions or junctions made with unconventional superconductors, or in spin-orbit weak links at large magnetic field - our group reported this in a recent paper (Ref. 54). They vanish in the time-reversal-symmetrical case (Ref. 6).

---

## Round 3 · List of Changes

In "Background: Current Phase Relations", we added a discussion about higher-order harmonics in SIS junctions.
In "Previous Work: Second Order Josephson Effect", we added a discussion about SQUIDs flux-biased to pi.
In "List of Results", we added sentences about the sign of the second harmonic.
In "Figure 1 Description", we added "The long effective length is likely due to large London penetration depth which is typical for thin film superconductors."
We added Fig. 6 and renamed "Non-identical negative and positive switching currents" to "Figure 6 Discussion". We added discussions about the inductive effect and the sign of the second harmonic term in this block.
In the supplementary, we added sections "Inductive SQUID" and "Sign of the second harmonic term" (including Figs. S5-S8). We added Fig. S9, which shows the zoomed-in data of Figs. 2(a) and 3(a).
Other changes.

---

## Editorial Decision

published